# ZC3H4 restricts non-coding transcription in human cells

**Chris Estell[1], Lee Davidson[1], Pieter C Steketee[2], Adam Monier[1], Steven West[1]***

[1]The Living Systems Institute, University of Exeter, Exeter, United Kingdom; [2]The Roslin Institute, Royal (Dick) School of Veterinary Studies, University of Edinburgh, Edinburgh, United Kingdom

**Abstract** The human genome encodes thousands of non-coding RNAs. Many of these terminate early and are then rapidly degraded, but how their transcription is restricted is poorly understood. In a screen for protein-coding gene transcriptional termination factors, we identified ZC3H4. Its depletion causes upregulation and extension of hundreds of unstable transcripts, particularly antisense RNAs and those transcribed from so-called super-enhancers. These loci are occupied by ZC3H4, suggesting that it directly functions in their transcription. Consistently, engineered tethering of ZC3H4 to reporter RNA promotes its degradation by the exosome. ZC3H4 is predominantly metazoan –interesting when considering its impact on enhancer RNAs that are less prominent in single-celled organisms. Finally, ZC3H4 loss causes a substantial reduction in cell proliferation, highlighting its overall importance. In summary, we identify ZC3H4 as playing an important role in restricting non-coding transcription in multicellular organisms.

*For correspondence:
S.West@exeter.ac.uk

**Competing interests:** The authors declare that no competing interests exist.

## Introduction

Most of the human genome can be transcribed by RNA polymerase II (Pol II). Among these transcripts are thousands of long non-coding RNAs, broadly classified as greater than ~200 nucleotides in length (*Kopp and Mendell, 2018*). They share some structural features with coding transcripts, but most of them are rapidly degraded by the exosome (*Davidson et al., 2019*; *Preker et al., 2008*; *Schlackow et al., 2017*). Their degradation is coincident with or shortly after transcriptional termination, which often occurs within a few kilobases (kb). The mechanisms for terminating non-coding transcription are poorly understood, especially by comparison with those operating at protein-coding genes.

Termination of protein-coding transcription is coupled to 3' end processing of pre-mRNA via cleavage at the polyadenylation signal (PAS) (*Eaton and West, 2020*). A PAS consists of an AAUAAA hexamer followed by a U/GU-rich region (*Proudfoot, 2011*). After assembly of a multi-protein processing complex, CPSF73 cleaves the nascent RNA and the Pol II-associated product is degraded 5'→3' by XRN2 to promote termination (*Eaton et al., 2018*; *Eaton et al., 2020*; *Fong et al., 2015*). The Pol II elongation complex is modified as it crosses the PAS, which facilitates its termination by XRN2 (*Cortazar et al., 2019*; *Eaton et al., 2020*). Depletion of XRN2 or CPSF73 causes read-through downstream of some long non-coding genes (*Eaton et al., 2020*). However, a substantial fraction of non-coding transcription is less sensitive to their depletion suggesting the use of alternative mechanisms.

The Integrator complex aids termination of many non-coding transcripts, with the archetypal example being snRNAs (*Baillat et al., 2005*; *Davidson et al., 2020*; *O'Reilly et al., 2014*). Integrator is also implicated in the termination of promoter upstream transcripts (PROMPTs) and enhancer RNAs (eRNAs) (*Beckedorff et al., 2020*; *Lai et al., 2015*; *Nojima et al., 2018*). The mechanism is analogous to that at protein-coding genes, driven by endonucleolytic cleavage by INTS11. However, INTS11 activity does not precede XRN2 degradation at snRNA genes (*Eaton et al., 2018*).

Moreover, while CPSF73 is indispensable for termination at protein-coding genes, there is evidence of redundant pathways at snRNA loci (*Davidson et al., 2020*). Indeed, CPSF and the cap binding complex-associated factor, ARS2, are both implicated in the termination of promoter-proximal transcription (*Iasillo et al., 2017*; *Nojima et al., 2015*).

A variety of processes attenuate transcription at protein-coding genes (*Kamieniarz-Gdula and Proudfoot, 2019*). Frequently, this is via premature cleavage and polyadenylation (PCPA) that can be controlled by U1 snRNA, CDK12, SCAF4/8, or PCF11 (*Dubbury et al., 2018*; *Gregersen et al., 2019*; *Kaida et al., 2010*; *Kamieniarz-Gdula et al., 2019*). PCPA is common on many genes since acute depletion of the exosome stabilises its predicted products in otherwise unmodified cells (*Chiu et al., 2018*; *Davidson et al., 2019*). Integrator activity also attenuates transcription at hundreds of protein-coding genes (*Elrod et al., 2019*; *Tatomer et al., 2019*).

A less-studied termination pathway at some intragenic non-coding regions is controlled by WDR82 and its associated factors (*Austenaa et al., 2015*). In mammals, WDR82 forms at least two complexes: one with the SETD1 histone methyl-transferase and another composed of protein-phosphatase 1 and its nuclear targeting subunit PNUTS (*Lee et al., 2010*; *van Nuland et al., 2013*). A version of the latter promotes transcriptional termination in trypanosomes (*Kieft et al., 2020*) and the budding yeast homologue of WDR82, Swd2, forms part of the APT (associated with Pta1) termination complex (*Nedea et al., 2003*). In murine cells, depletion of either WDR82, PNUTS, or SET1 causes non-coding transcriptional termination defects (*Austenaa et al., 2015*). Notably, PNUTS/PP1 is also implicated in the canonical termination pathway at protein-coding genes where its dephosphorylation of SPT5 causes deceleration of Pol II beyond the PAS (*Cortazar et al., 2019*; *Eaton et al., 2020*).

Here, we performed a proteomic screen for new termination factors by searching for proteins that bind to Pol II complexes in a manner that depends on PAS recognition by CPSF30. This uncovered ZC3H4, a metazoan zinc finger-containing factor without a characterised function in transcription. Because of the nature of our screen, we anticipated a role for ZC3H4 in 3' end formation; however, its effects on this process are mild and apply to a small number of genes. Its main function is to restrict non-coding transcription, especially of PROMPT and eRNA transcripts, which can be extended by hundreds of kb when ZC3H4 is depleted. ZC3H4 interacts with WDR82, the depletion of which causes similar defects. Tethered function assays show that ZC3H4 recruitment is sufficient to restrict transcription and cause RNA degradation by the exosome. In sum, we reveal ZC3H4 as a hitherto unknown terminator of promoter-proximal transcription with particular relevance at non-coding loci.

## Results

### The effect of CPSF30 depletion on the Pol II-proximal proteome

The first step of PAS recognition involves the binding of CPSF30 to the AAUAAA signal (*Chan et al., 2014*; *Clerici et al., 2018*; *Sun et al., 2018*). We reasoned that elimination of CPSF30 would impede PAS-dependent remodelling of Pol II elongation complexes and cause the retention or exclusion of potentially undiscovered transcriptional termination factors. We used CRISPR/Cas9 genome editing to tag *CPSF30* with a mini auxin-inducible degron (mAID) (*Figure 1A*). The integration was performed in HCT116 cells where we had previously introduced the plant F-box gene, *TIR1*, required for the AID system to work (*Eaton et al., 2018*; *Natsume et al., 2016*). CPSF30-mAID is eliminated by 3 hr of indol-3-acetic acid (auxin/IAA) treatment (*Figure 1B*). This results in profound and general transcriptional read-through downstream of protein-coding genes (*Figure 1C* and *Figure 1—figure supplement 1A*) demonstrating widespread impairment of PAS function.

To identify Pol II interactions sensitive to CPSF30, we further modified *CPSF30-mAID* cells to homozygously tag the largest subunit of Pol II, Rpb1, with mini(m)-Turbo (*Figure 1D* and *Figure 1—figure supplement 1B*). mTurbo is an engineered ligase that biotinylates proximal proteins when cells are exposed to biotin (*Branon et al., 2018*). This occurs within minutes of biotin addition to culture media, which is advantageous for analysing dynamic proteins such as Pol II. We chose this approach rather than immunoprecipitation (IP) because it allows isolation of weak/transient interactions (potentially disrupted during conventional IP) and may identify relevant proximal proteins that

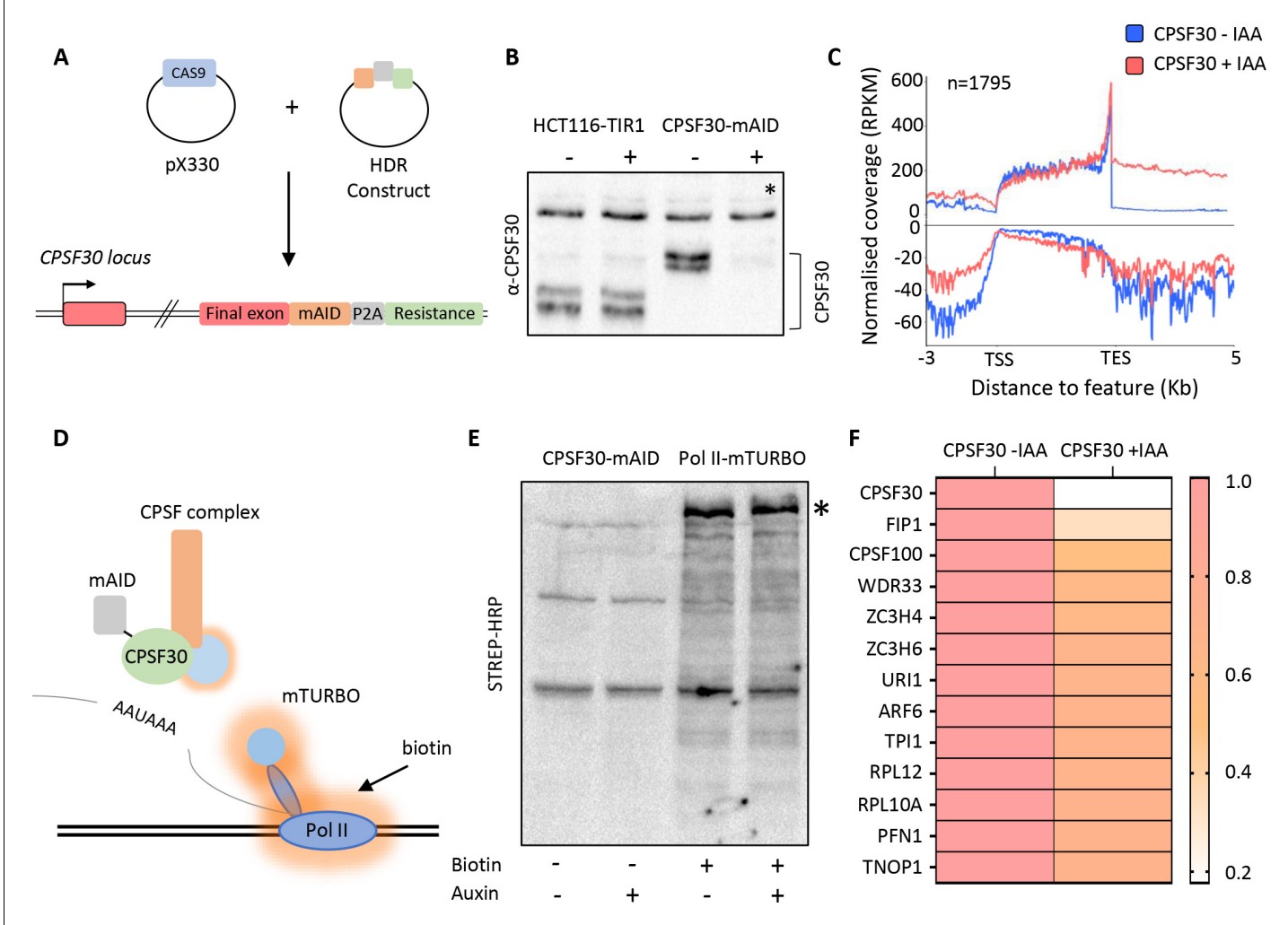

**Figure 1.** Proximity labelling of CPSF30-sensitive Pol II interactions by mTurbo. (a) Schematic of the strategy used to tag CPSF30 with the mini auxin-inducible degron (mAID). Guide RNA-expressing Cas9 plasmid and homology-directed repair (HDR) plasmids are shown and the resulting modification to *CPSF30* is represented with each inserted element labelled. (b) Western blot demonstrating CPSF30 depletion. Parental HCT116-TIR1, or *CPSF30-mAID* cells, were treated ±auxin for 3 hr, then blotted. CPSF30 protein is indicated together with a non-specific product, marked by an asterisk, used as a proxy for protein loading. (c) Metagene analysis of 1795 protein-coding genes demonstrating increased downstream transcription, derived from sequencing nuclear RNA, following auxin treatment (3 hr) of *CPSF30-mAID* cells. TSS = transcription start site, TES = transcription end site (PAS), read-through signal is normalised against gene body. RPKM is reads per kilobase of transcript, per million mapped reads. Positive and negative signals represent sense and antisense reads, respectively. (d) Schematic of our strategy to identify new factors involved in transcription termination. *CPSF30-mAID* cells were edited to express Rpb1-mTurbo (blue circle on Pol II). The addition of biotin induces mTurbo-mediated biotinylation (orange haze) of factors proximal to Pol II. CPSF complex is shown as an example of what might be captured by this experiment. (e) Western blot showing streptavidin horseradish peroxidase (HRP) probing of extracts from *CPSF30-mAID: RPB1-mTurbo* cells. Prior treatment with auxin (3 hr)/biotin (10 min) is indicated. The high molecular weight species in the +biotin samples corresponds in size to Rpb1-mTurbo (*). (f) Heat map detailing proteins with the largest decrease in Pol II interaction. Data underpinning heat map are from mass spectrometry analysis of streptavidin sequestered peptides (±CPSF30) performed in triplicate. Labelling was for 10 min.

The online version of this article includes the following figure supplement(s) for figure 1:

**Figure supplement 1.** Validation of the CPSF30 transcriptional read-through defect and of tagging *RPB1* with mTurbo.

**Figure supplement 2.** Predicted structures and interactors of ZC3H4 and ZC3H6.

**Figure supplement 3.** Phylogenetic analysis of ZC3H4 and ZC3H6.

do not interact with Pol II directly. Importantly, CPSF30-mAID depletion still induced strong read-through in this cell line (*Figure 1—figure supplement 1C*).

*CPSF30-mAID:RPB1-mTurbo* cells were exposed to biotin before western blotting with streptavidin horseradish peroxidase (HRP). This revealed multiple bands with a prominent one corresponding in size to Rpb1-mTurbo and indicating the biotinylation of its proximal proteome (*Figure 1E*). A small number of endogenously biotinylated factors were observed in the absence of biotin. Biotin-exposed samples were subject to tandem mass tagging (TMT) with mass spectrometry. We focused on proteins with reduced abundance after auxin treatment (*Supplementary file 1*). The factor most depleted was CPSF30, confirming that its auxin-dependent depletion is reflected in the data (*Figure 1F*). As expected, Rpb1 was the most abundant factor in all samples consistent with its self-biotinylation seen by western blotting. After CPSF30, the most depleted factors were Fip1, CPSF100, and WDR33, which are in the CPSF complex. Otherwise, surprisingly few proteins showed reduced signal following auxin treatment. This implies that the major effect of CPSF30 depletion on the Pol II-proximal interactome is to prevent the recruitment/assembly of the CPSF complex.

## ZC3H4 is a candidate transcription termination factor that is metazoan-enriched

Two poorly characterised factors, ZC3H4 and ZC3H6, were the next most depleted. They contain CCCH zinc finger motifs flanked by intrinsically disordered regions (*Figure 1—figure supplement 2A*). Their potential relationship to canonical 3' end formation factors is suggested via known/predicted protein-protein interactions that are collated by the STRING database (*Jensen et al., 2009*; *Figure 1—figure supplement 2B*). ZC3H4 is also co-regulated with mRNA processing factors suggesting a role in RNA biogenesis (*Figure 1—figure supplement 2C*; *Kustatscher et al., 2019*). Although little is reported on ZC3H4, two independent studies uncovered it as an interaction partner of WDR82 using mass spectrometry (*Lee et al., 2010*; *van Nuland et al., 2013*). WDR82 plays a key role in transcriptional termination in yeast, trypanosomes, and mice (*Austenaa et al., 2015*; *Kieft et al., 2020*; *Nedea et al., 2003*). To verify this interaction, we tagged ZC3H4 with GFP and performed a 'GFP trap' whereby ZC3H4-GFP is captured from whole cell lysates using GFP nanobody-coupled beads (*Figure 1—figure supplement 2D*). WDR82 robustly co-precipitated with ZC3H4-GFP, confirming them as interacting partners. Although WDR82 is conserved between human and budding yeast, our phylogenetic analysis suggested that ZC3H4 and ZC3H6 are largely restricted to metazoans and are paralogues (*Figure 1—figure supplement 3A and B*).

## ZC3H4 restricts non-coding transcription events

To assess any function of ZC3H4 and/or ZC3H6 in RNA biogenesis, we depleted either or both from HCT116 cells using RNA interference (RNAi) (*Figure 2—figure supplement 1A*), then deep sequenced nuclear transcripts. Comparison of these datasets shows that ZC3H4 loss has a more noticeable impact than ZC3H6 depletion (*Figure 2—figure supplement 1B*). Specifically, ZC3H6 depleted samples are more similar to control than those deriving from ZC3H4 loss and ZC3H4/ZC3H6 co-depletion resembles a knockdown of just ZC3H4. This was also evident from closer inspection of the data (*Figure 2—figure supplement 1C*), supporting the phylogenetic prediction of their separate functions. Accordingly, subsequent analyses focus on ZC3H4.

Due to its links with CPSF30 and WDR82, we anticipated that ZC3H4 might affect transcriptional termination. We first checked protein-coding genes and found a small number with longer read-through beyond the PAS when ZC3H4 is depleted (*Figure 2A*). However, broader analysis suggests that this is not widespread and far fewer genes exhibit increased read-through following ZC3H4 loss compared to when CPSF30 is absent (*Figure 2B* and *Figure 2—figure supplement 2A–D*). Interestingly, the metagene in *Figure 2B* revealed slightly more signal antisense of promoters when ZC3H4 is depleted. This indicates an effect on non-coding RNA, which is interesting in light of a previously described function for WDR82 in restricting intragenic transcription (*Austenaa et al., 2015*). These PROMPT transcripts are normally rapidly degraded 3'→5' by the exosome (*Preker et al., 2008*). *Figure 2C* shows an example PROMPT, upstream of *MYC*, which is undetectable in control siRNA-treated cells, but abundant following ZC3H4 depletion. Loss of ZC3H4 also leads to the extension of this transcript by more than 100 kilobases. This is made clearer by comparing the loss of ZC3H4 to AID-mediated depletion of the catalytic exosome (DIS3) (*Davidson et al., 2019*). DIS3 depletion

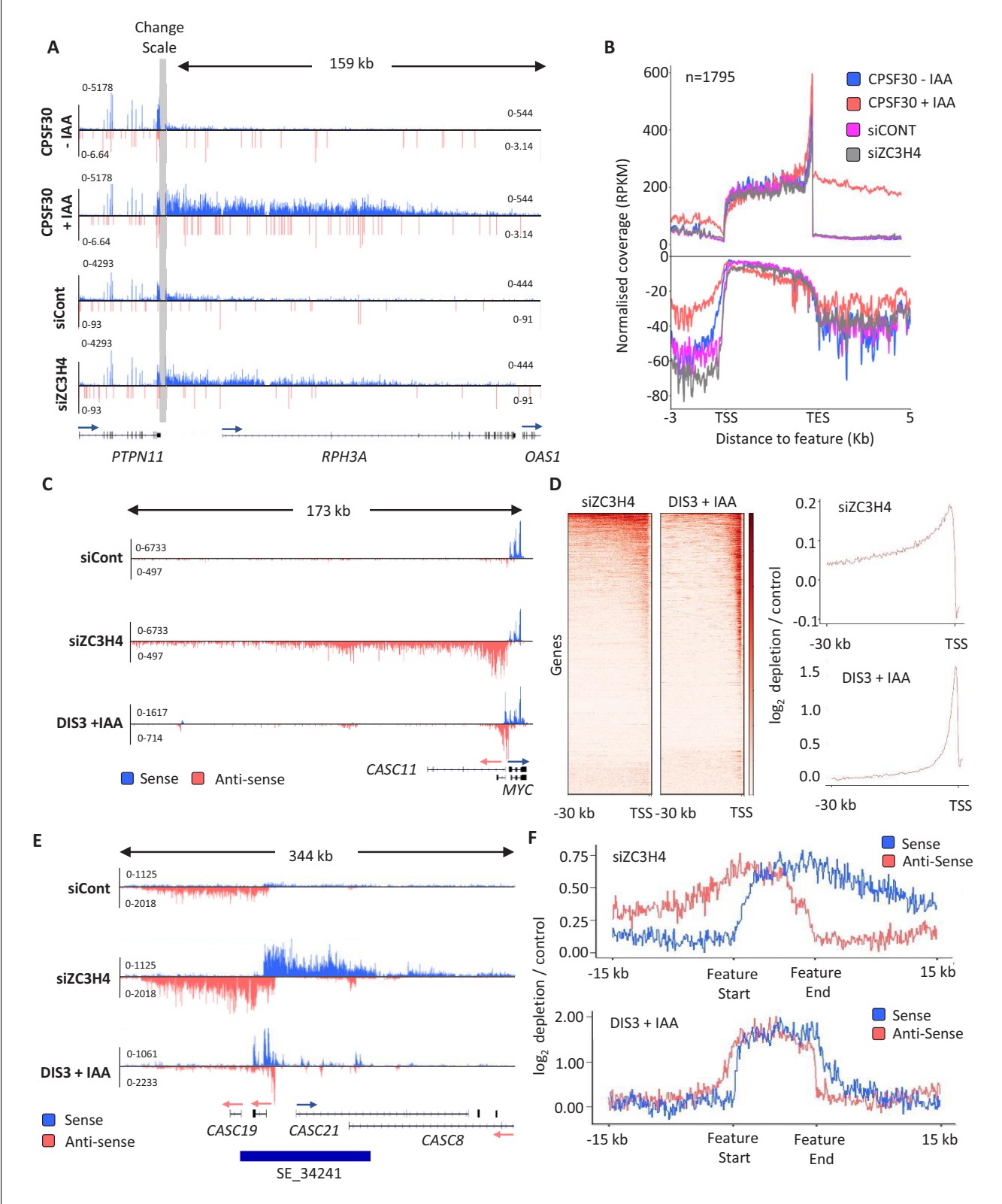

**Figure 2.** ZC3H4 depletion stabilises unproductive transcripts. (a) Integrative Genomics Viewer (IGV) track of the transcription read-through defect at *PTPN11* following CPSF30 or ZC3H4 depletion. Blue and red tracks indicate sense/antisense transcripts respectively, grey bar indicates a change in y-axis scale so that comparatively weaker read-through signals can be visualised next to the gene body (left scale for upstream of TES; right for downstream). Y-axis scale is RPKM. (b) Metagene comparison of transcription upstream, across, and downstream of, protein-coding genes in nuclear

*Figure 2 continued on next page*

*Figure 2 continued*

RNA from *CPSF30-mAID* cells treated or not with auxin and from HCT116 cells transfected with control or ZC3H4 siRNAs. CPSF30 traces are from the same samples presented in *Figure 1C*. Positive and negative signals represent sense and antisense reads, respectively. (c) IGV track view of transcription at the *MYC* PROMPT in RNA-seq samples obtained from control or ZC3H4 siRNA-treated HCT116 cells. We also show a track from HCT116 cells acutely depleted of DIS3-AID (DIS3 + IAA) (*Davidson et al., 2019*) to highlight the normal extent of this unstable transcript. Y-axis scale is RPKM. (d) Log2 fold change of siZC3H4 vs. siControl or DIS3+ vs. - auxin for RNA upstream of 6057 non-neighbouring, actively transcribed genes, plotted as heat maps. Line graphs are an XY depiction of heat map data. Log2 fold changes are smaller in siZC3H4 samples versus DIS3 depletion because this is an average of all genes in the heat map, a smaller fraction of which are affected by ZC3H4. (e) IGV plot of a known SE upstream of *MYC* (the location is shown by blue bar under trace). Samples are shown from HCT116 cells treated with control or ZC3H4 siRNAs as well as *DIS3-AID* cells treated with auxin (the latter from *Davidson et al., 2019*) to show the normal extent of unstable eRNAs over this region. Y-axis scale is RPKM. (f) Log2 fold change of RNA signal for siZC3H4 vs. siControl or DIS3+ vs. - auxin for 111 SEs. The bed file detailing super-enhancer coordinates in HCT116 cells was taken from dbSUPER.org.

The online version of this article includes the following figure supplement(s) for figure 2:

**Figure supplement 1.** Comparison of ZC3H4 or ZC3H6 depletion, and their co-depletion, via RNA-seq.
**Figure supplement 2.** Comparison of CPSF30 and ZC3H4 effects on transcriptional read-through.

stabilises the usual extent of PROMPT RNA, which is much shorter than when ZC3H4 is absent. Importantly, meta-analysis reveals similar effects at many other PROMPTs (*Figure 2D*). These data strongly suggest that PROMPT transcripts are stabilised and extended in the absence of ZC3H4, presumably because its normal function restricts their transcription.

The finding that PROMPTs are affected by ZC3H4 suggested a role in the transcription/metabolism of antisense/non-coding RNAs. We therefore extended our search for potential ZC3H4 regulated transcription to enhancer regions since they also produce short RNAs that are degraded by the exosome (*Andersson et al., 2014*). eRNAs can be found in isolation and in clusters called super-enhancers (SEs) (*Pott and Lieb, 2015*). SEs are thought to be important for controlling key developmental genes with strong relevance to disease (*Hnisz et al., 2013*). ZC3H4 depletion has a clear effect over SE regions exemplified by the *MYC* SE where upregulation and extension of eRNAs is obvious (*Figure 2E*). Acute depletion of DIS3 illustrates the normally restricted range of individual eRNAs within the cluster. This effect is general for other SEs as demonstrated by the metaplots in *Figure 2F*. We also checked the effect of CPSF30 depletion on example PROMPT and SE transcription, which are very modest and consistent with the lack of antisense effect seen by metagene in *Figure 1C* (*Figure 2—figure supplement 2E*). Consistently, PROMPTs susceptible to ZC3H4 were not enriched in PASs compared to those unaffected by it and harbour a slightly lower density (*Figure 2—figure supplement 2F*). Overall, these data strongly suggest that ZC3H4 is important for regulating transcription across many PROMPTs and SEs.

## Comparison of ZC3H4 and Integrator effects

ZC3H4 has some functions in common with the Integrator complex. This is a metazoan-specific assembly with regulatory functions at non-coding loci (*Lai et al., 2015*; *Mendoza-Figueroa et al., 2020*; *Nojima et al., 2018*). We previously sequenced chromatin-associated RNA derived from HCT116 cells RNAi depleted of the Integrator backbone component INTS1 (*Davidson et al., 2020*). Chromatin-associated RNA is purified via urea/detergent extraction and is enriched in nascent RNAs (*Wuarin and Schibler, 1994*). Metagene analysis of this data at protein-coding genes shows a mild effect of Integrator depletion over PROMPT regions (*Figure 3A*). It also reveals an accumulation of promoter-proximal RNAs in the coding direction consistent with a recent report on its function as an attenuator of protein-coding transcription (*Lykke-Andersen et al., 2020*). Because of this function, Integrator depletion can lead to increased expression of a subset of mRNAs (*Elrod et al., 2019*; *Lykke-Andersen et al., 2020*; *Tatomer et al., 2019*). *HAP1* is an example of a gene where this is seen (*Figure 3B*). Similarly, we saw evidence for increased mRNA expression on some genes when ZC3H4 was depleted (*Figure 3C*). Interestingly, these two genes are selectively effected by Integrator or ZC3H4, respectively, and additional examples of this are shown in *Figure 3—figure supplement 1A*. Bioinformatic analysis revealed around 1000 genes affected by INTS1 or ZC3H4 depletion with little overlap between the two conditions (*Figure 3D*, *Supplementary file 4*). Indeed, analysis of recently published metabolically labelled RNA-seq data from HeLa cells depleted of the catalytic Integrator subunit or ZC3H4 reveals many upregulated mRNAs – also with minimal overlap

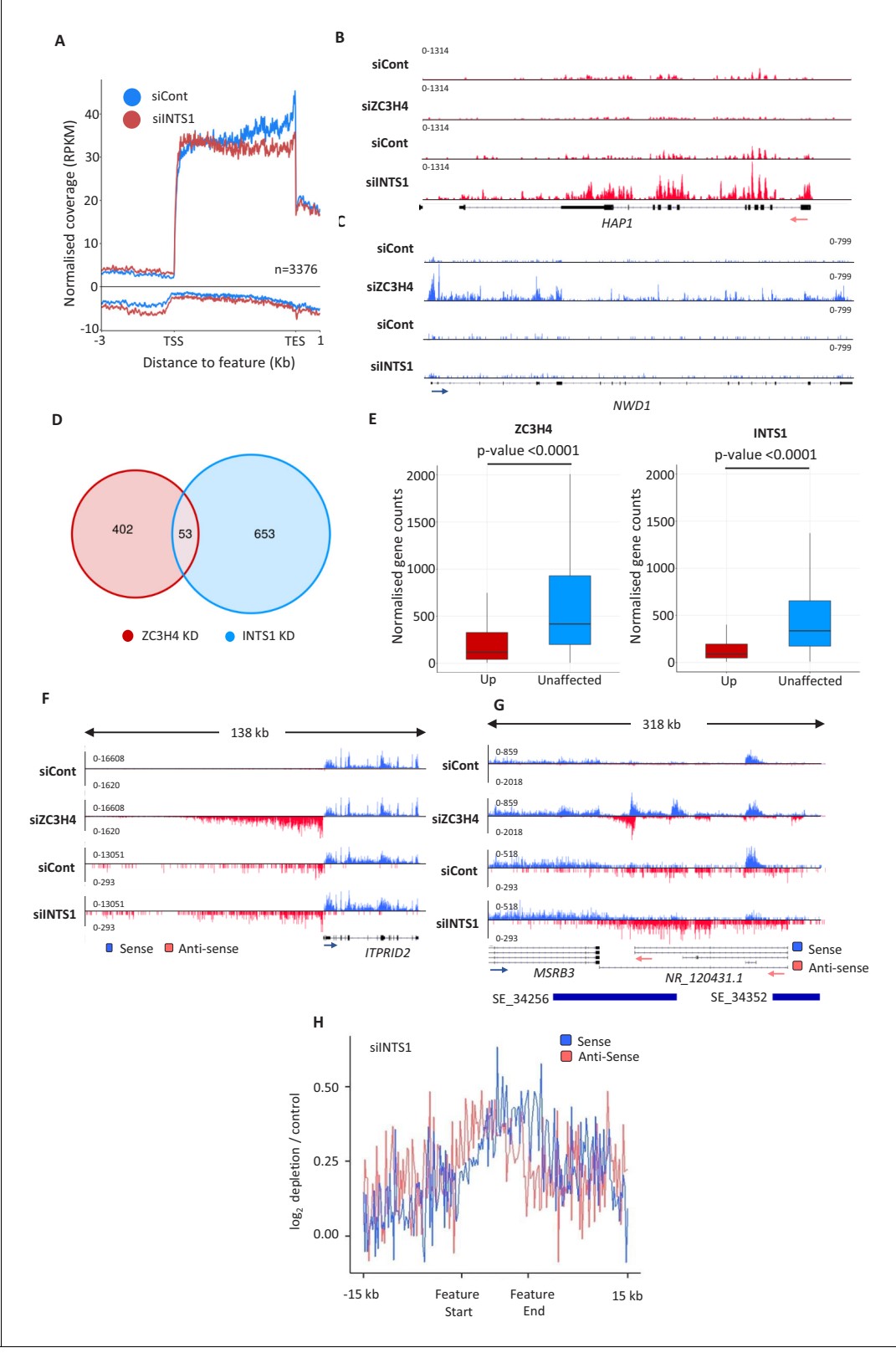

**Figure 3.** Comparison of ZC3H4 and Integrator effects. (**a**) Metagene analysis of chromatin-associated RNA-seq performed on cells treated with control or INTS1-specific siRNA. The plot shows signals upstream, across, and downstream of protein-coding genes. Y-axis scale is RPKM. Positive and negative values represent sense and antisense reads, respectively. (**b, c**) IGV traces of *HAP1* and *NWD1* genes derived from chromatin-associated RNA-seq in control and INTS1 siRNA treated samples and nuclear RNA-seq from control or ZC3H4 siRNA treatment. *NWD1* transcripts are affected by

*Figure 3 continued on next page*

Figure 3 continued

ZC3H4 but not INTS1, whereas the opposite is true for *HAP1* RNAs. Y-axes scales are RPKM. (**d**) Venn diagram showing the number of mRNAs upregulated ≥2-fold, padj ≤0.05 following ZC3H4 depletion versus INTS1 loss and the overlap between the two sets. Genes that showed increased expression due to transcription read-through from an upstream gene were filtered by assessing coverage over a 1 kb region preceding the TSS, relative to untreated cells. (**e**) Graphs demonstrating the expression level of mRNA transcripts upregulated (log2FC >1) following ZC3H4 or INTS1 depletion by comparison with transcripts unaffected by loss of either factor. Y-axis shows normalised gene counts (i.e. expression level). (**f**) Comparison of chromatin-associated RNA-seq in control and INTS1 siRNA treated samples with nuclear RNA-seq derived from control or ZC3H4 siRNA treatment. The *ITPRID2* PROMPT is displayed and y-axes are RPKM (note the different scales between ZC3H4 and INTS1 samples). (**g**) Comparison of chromatin-associated RNA-seq in control and INTS1 siRNA treated samples with nuclear RNA-seq derived from control or ZC3H4 siRNA treatment. The *MSRB3* SE is displayed and y-axes are RPKM (note the different scales between INTS1 and ZC3H4 samples). (**h**) Metaplot of RNA-seq profile over super-enhancers following INTS1 depletion (log2 fold depletion/control over 111 super-enhancer as line graphs). The bed file detailing super-enhancer coordinates in HCT116 cells was taken from dbSUPER.org. RPKM = reads per kilobase of transcript, per million mapped reads, TSS = transcription start site.

The online version of this article includes the following figure supplement(s) for figure 3:

**Figure supplement 1.** Comparison of ZC3H4 and INTS1 depletion on mRNA and PROMPT transcripts.

(*Austenaa et al., 2021*; *Lykke-Andersen et al., 2020*; *Figure 3—figure supplement 1B*). When searching for characteristics of these targets in our own RNA-seq data, we found that transcripts upregulated following either ZC3H4 or INTS1 loss are normally expressed at lower levels than those from unaffected genes (*Figure 3E*). This is consistent with the idea that they are subject to repression by these two factors under these experimental conditions.

The most prominent effects of ZC3H4 were observed at PROMPT and SE regions where, again, Integrator is implicated (*Lai et al., 2015*; *Nojima et al., 2018*). Where ZC3H4 effects are evident over PROMPT regions, they are generally more substantial than those seen after Integrator loss, exemplified by the *ITPRID2* PROMPT in *Figure 3F* and via meta-analyses (*Figure 3—figure supplement 1C and D*). At SEs, ZC3H4 depletion generally results in a greater stabilisation and elongation of eRNA, compared to INTS1 knockdown, exemplified at the *MSRB3* SE (*Figure 3G*). Meta-analysis confirms less effect of INTS1 depletion at SEs versus the impact of ZC3H4 (compare *Figures 3H* and *2F*). We note that these INTS1 data are on chromatin-associated RNA whereas ZC3H4 images are obtained from nuclear RNA. However, as chromatin-associated RNA is more enriched in nascent transcripts, this would be expected to capture more extended non-coding transcription and not less as is the case here. Moreover, previously published analyses of Integrator effects on transcription do not report the long extended non-coding (PROMPT/eRNA) transcripts that we observe when ZC3H4 is depleted (*Beckedorff et al., 2020*; *Lykke-Andersen et al., 2020*).

## Rapid ZC3H4 depletion and re-expression confirms the functions found by RNA-seq

ZC3H4 RNAi suggests its widespread involvement in non-coding RNA synthesis and the regulation of a subset of protein-coding transcripts. However, RNAi depletion was performed using a 72 hr protocol and might result in indirect or compensatory effects. To assess whether these effects are a more direct consequence of ZC3H4 loss, we engineered HCT116 cells for its rapid and inducible depletion. CRISPR/Cas9 was used to tag *ZC3H4* with an *E. coli* derived DHFR degron preceded by 3xHA epitopes (*Figure 4A*; *Sheridan and Bentley, 2016*). In this system, cells are maintained in trimethoprim (TMP) to stabilise the degron, removal of which causes protein depletion. Western blotting demonstrates homozygous tagging of *ZC3H4* and that ZC3H4-DHFR is depleted following TMP removal (*Figure 4B*). Depletion was complete after overnight growth without TMP but substantial protein loss was already observed after 4 hr allowing us to assess the consequences of more rapid ZC3H4 depletion.

TMP-mediated depletion can also be reversed by its re-administration facilitating a test of whether ZC3H4 effects are reversed by its re-appearance. The western blot in *Figure 4C* illustrates this by showing that TMP withdrawal depletes ZC3H4-DHFR, which re-appears following 4 hr TMP addition. To ask whether ZC3H4 effects are an immediate consequence of its loss and if they are reversed following its re-appearance, RNA was isolated from the three conditions shown in the western blot. This was analysed by quantitative reverse transcription and PCR (qRT-PCR) to assess the levels of extended PROMPT (*HMGA2, ITPRID2*) and SE (*MSRB3, DLGAP1*) RNAs (*Figure 4D*). All were increased following ZC3H4 loss, suggesting that the effects that we observed by RNAi are not

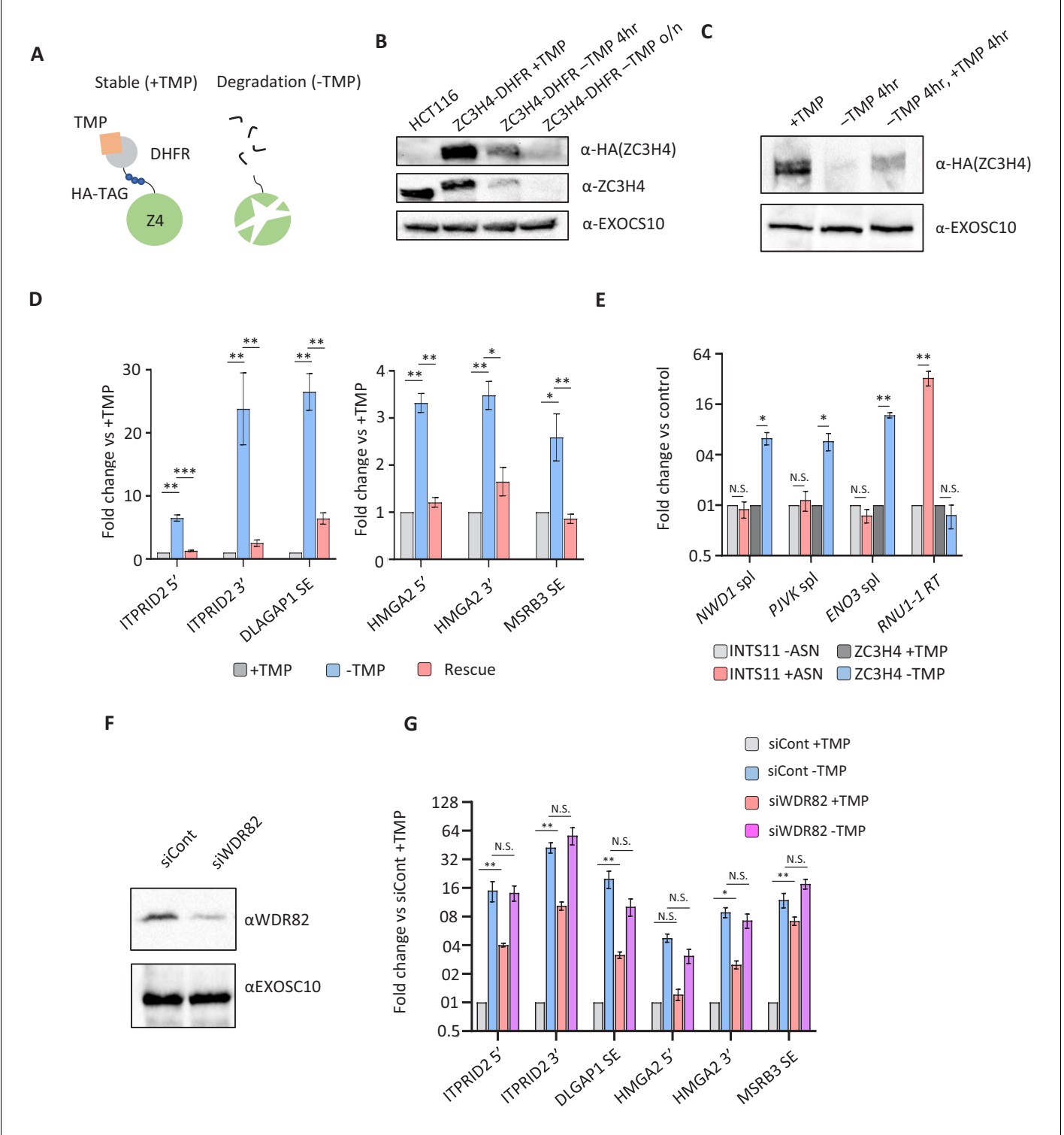

**Figure 4.** Transcriptional dysregulation following acute ZC3H4 loss. (a) Schematic detailing how the dihydrofolate reductase (DHFR) degron works. *E. coli* dihydrofolate reductase (DHFR) is fused to the C-terminus of ZC3H4, which is stabilised by trimethoprim (TMP). When TMP is removed, ZC3H4-DHFR is degraded. (b) Western blot of HCT116 parental and HCT116 *ZC3H4-DHFR* cells ± TMP. TMP was withdrawn for 4 hr or overnight (o/n), EXOCS10 is used as a loading control, αHA recognises a HA peptide before the DHFR tag, while αZC3H4 recognises native protein. (c) Western blot of *ZC3H4-DHFR* cells grown under the following conditions: +TMP, –TMP (4 hr), –TMP (4 hr) followed by +TMP (4 hr). ZC3H4-DHFR is detected using αHA, and EXOSC10 is shown as a loading control. (d) qRT-PCR analysis of PROMPT and SE transcripts in *ZC3H4-DHFR* cells grown under the conditions represented in (c) (rescue refers to –TMP then +TMP for re-establishing ZC3H4). Graph shows fold change versus +TMP following normalisation to

*Figure 4 continued on next page*

Figure 4 continued

spliced actin. N = 3. Error bars are SEM. *, **, and *** denote p<0.05, 0.01, and 0.001, respectively. ITPRID2 5' 3' primers are at approximately −500 bp and −7 kb relative to its TSS. HMGA2 5' and 3' primers are at approximately −1.8 kb and −7.1 kb relative to its TSS. (e) qRT-PCR analysis of spliced PJVK, ENO3 and NWD1 mRNAs and RNU1-1 read-through (RT) in *ZC3H4-DHFR* cells grown with or without (4 hr) TMP and *INTS11-SMASh* cells grown with or without asunaprevir (ASN; 36 hr to deplete INTS11-SMASh). Graph shows fold change versus control (+TMP for ZC3H4-DHFR samples and –ASN for INTS11-SMASh samples), following normalisation to spliced actin. N = 3. Error bars are SEM. * and ** denote p<0.05 and 0.01, respectively. (f) Western blot of extracts derived from HCT116 cells transfected with control or WDR82-specific siRNAs. The blot shows WDR82 and, as a loading control, EXOSC10. (g) qRT-PCR of PROMPT and SE transcripts in *ZC3H4-DHFR* cells transfected with control or WDR82 siRNAs before withdrawal, or not, of TMP (14 hr). Graph shows fold change by comparison with control siRNA transfected *ZC3H4-DHFR* cells maintained in TMP following normalisation to spliced actin transcripts. N = 3. Error bars are SEM. * and ** denote p<0.05 and 0.01, respectively. PROMPT = promoter upstream transcript.

The online version of this article includes the following figure supplement(s) for figure 4:

**Figure supplement 1.** ZC3H4, PNUTS and SETD1A/B effects on PROMPT and SE transcripts/transcription.

due to compensatory pathways. Although 4 hr TMP re-administration does not restore ZC3H4 to full levels, it was sufficient to reverse the effects of its depletion at all tested amplicons. The timescale over which the effect can be reversed suggests that transcripts induced by ZC3H4 loss remain relatively unstable. Rapid ZC3H4 depletion also confirmed the prediction, from our RNA-seq, that the extended PROMPT transcripts result from the aberrant transcription of these loci (*Figure 4—figure supplement 1A and B*).

Another key observation from our nuclear RNA-seq was the potential for ZC3H4 to restrict the levels of a subset of protein-coding transcripts. The long-term nature of RNAi and its detection via nuclear RNA-seq means that it could be an indirect or post-transcriptional effect, respectively. To test whether mRNA upregulation is an immediate and transcriptional response to ZC3H4 loss, we isolated chromatin-associated RNA from *ZC3H4-DHFR* cells grown with or without TMP for 4 hr. To additionally confirm their specificity to ZC3H4 (vs. Integrator), we also depleted the catalytic Integrator subunit utilising our previously engineered cell line in which INTS11 is tagged with a small molecule assisted shut-off module (*Chung et al., 2015*; *Davidson et al., 2020*). qRT-PCR was used to detect three transcripts (NWD1, ENO3, and PJVK) that were upregulated by ZC3H4 loss but not Integrator depletion. Spliced versions of all three were increased after 4 hr of ZC3H4 depletion, but unaffected by loss of the catalytic Integrator subunit INTS11 (*Figure 4E*). The effectiveness of INTS11 depletion is illustrated by the substantial increase in U1 snRNA read-through RNA in its absence. This demonstrates that some mRNAs are immediately and selectively upregulated following ZC3H4 loss.

We next asked whether the ZC3H4 interactor, WDR82, impacts the levels of PROMPT and SE transcripts. Accordingly, *ZC3H4-DHFR* cells were treated with control or WDR82-specific siRNAs (*Figure 4F*). We also co-depleted ZC3H4 and WDR82 by removing TMP from cells first transfected with WDR82 siRNAs. WDR82 depletion enhanced the level of all tested transcripts suggesting that it functionally overlaps with ZC3H4 (*Figure 4G*). There was no synergistic effect of their co-depletion, implying that WDR82 and ZC3H4 do not act redundantly at the tested loci. WDR82 is found in complexes containing protein phosphatase 1 (PP1) and the SETD1A/B methyl transferases (*Lee et al., 2010*; *van Nuland et al., 2013*). We found that the former but not the latter is implicated in the stability of the non-coding transcripts selected for this experiment (*Figure 4—figure supplement 1C–E*).

## ZC3H4 occupies a broad region at a subset of promoters

We have demonstrated that depletion of ZC3H4 causes widespread defects in non-coding transcription and supresses a subset of protein-coding RNAs. As these effects are seen following rapid ZC3H4 depletion, we hypothesised that they may be directly mediated by its recruitment to relevant loci. Consistently, its capture in our mTurbo experiment supports its proximity to chromatin, and the presence of CCCH zinc finger domains predicts nucleic acid binding capability. Therefore, its genomic occupancy was globally investigated by performing ZC3H4 chromatin immunoprecipitation and sequencing (ChIP-seq) alongside that of Pol II.

ZC3H4 occupies genes with binding broadly resembling that of Pol II and showing the greatest enrichment over promoter regions (*Figure 5A*). However, many genes that are occupied by Pol II do

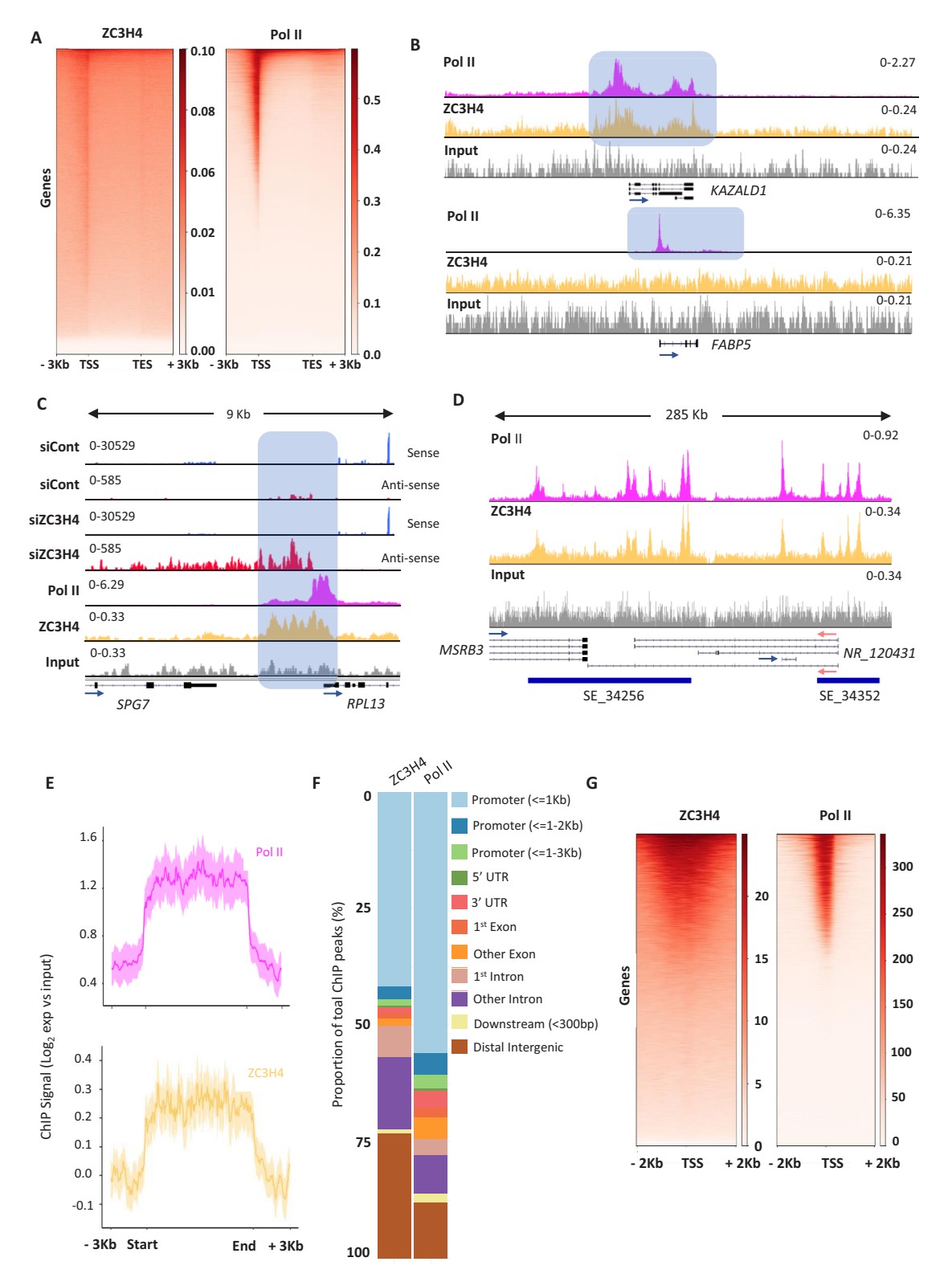

**Figure 5.** ZC3H4 occupies regions where transcription is affected by its absence. (a) ZC3H4 ChIP profile over protein-coding genes is similar to Pol II. Heat map representation of ZC3H4 and Pol II ChIP-seq occupancy over the gene body ±3 kb. (b) ZC3H4 occupies fewer promoters than Pol II. IGV track view of ZC3H4 and Pol II occupancy over *KAZALD1* and *FABP5* genes, Pol II is present at both genes, while ZC3H4 is only present at *KAZALD1*. Scale is counts per million (CPM). Shaded blue box shows peak of Pol II and ZC3H4 at *KAZALD1* and of Pol II over *FABP5*. (c) RNA-seq (HCT116 cells treated

*Figure 5 continued on next page*

*Figure 5 continued*

with control or ZC3H4 siRNA) and ChIP-seq (Pol II, ZC3H4 and input) profiles at *RPL13*. ZC3H4 occupancy is focused more on the PROMPT transcript region (blue box) than the TSS where, in contrast, the Pol II signal is maximal. RNA-seq scale is RPKM and ChIP-seq is CPM. (d) ZC3H4 ChIP occupancy mirrors Pol II at super-enhancers. IGV track view of ZC3H4 and Pol II occupancy over the SE at the *MSRB3* locus. HCT116 super-enhancer gene track is from dbSUPER and depicted as blue bars. (e) Log2 fold change of ZC3H4 and Pol II vs. input at SEs shown as a line graph. Halo denotes 95% confidence level. (f) ChIPseeker analysis of peak distribution of ZC3H4 and Pol II. Occupancy regions are colour-coded and the number of ChIP peaks expressed as a proportion of 100%. (g) Heat map showing Pol II and ZC3H4 ChIP occupancy in HEPG2 cells obtained via the ENCODE consortium. Occupancy ±2 kb of the TSS is shown. RPKM = reads per kilobase of transcript, per million mapped reads, ChIP-seq = chromatin immunoprecipitation and sequencing, CPM = counts per million, TSS = transcription start site.

The online version of this article includes the following figure supplement(s) for figure 5:

**Figure supplement 1.** ZC3H4 RNA binding in HCT116 cells and ChIP-seq comparison of its occupancy of super-enhancers in HCT116 and HEPG2 cells.

not recruit ZC3H4 (*Figure 5B*). This might result from low affinity of the ZC3H4 antibody or that its recruitment to chromatin is bridged since ZC3H4 also directly crosslinks to RNA in cells (*Figure 5— figure supplement 1A*). The differential occupancy of genes by ZC3H4 is consistent with the selective effects of its depletion. Interestingly, ZC3H4 occupies a broader promoter region than Pol II, suggesting that its function is not restricted to the precise transcriptional start site. The width of this peak often corresponds to the normal extent of PROMPT and eRNA transcription, which is elongated in its absence. *RPL13* is shown as an example of recruitment of ZC3H4 upstream of the promoter, where its loss causes stabilisation and extension of the antisense transcript (*Figure 5C*). ZC3H4 is also strongly recruited to SEs consistent with the RNA effects observed on them following its loss (*Figure 5D*). This is exemplified by the *MSRB3* region and generalised by metaplots in *Figure 5E*. Although our analyses of eRNA and PROMPTs were guided by our RNA-seq findings, an unbiased search for peaks of ZC3H4 and Pol II signal confirmed proportionally greater ZC3H4 occupancy at distal intergenic regions (encompassing SEs) (*Figure 5F*).

Overall, the HCT116 ChIP-seq demonstrates direct recruitment of ZC3H4 to potential targets. One mentioned caveat is the low ChIP efficiency of the ZC3H4 antibody; however, a ZC3H4 ChIP-seq experiment was recently made available by the ENCODE consortium (*Partridge et al., 2020*). This used a flag-tagged construct and was performed in HEPG2 cells allowing a comparison of our data to that obtained with a high-affinity antibody and in different cells. Consistent with our findings, flag-ZC3H4 occupies a subset of Pol II-bound regions and shows broader distribution than Pol II around promoters (*Figure 5G*). Although HEPG2 cells express fewer SEs than HCT116 cells, the transcribed *DLGAP1* example confirms its occupancy of these regions in both cell types (*Figure 5— figure supplement 1B*). In contrast, the *MYC* SE is only expressed in HCT116 cells and is not occupied by ZC3H4 in HEPG2 cells. In further agreement with our data, bioinformatics assignment of flag-ZC3H4 binding sites yielded 'promoter and enhancer-like' as the most enriched terms (*Partridge et al., 2020*).

## Engineered recruitment of ZC3H4 suppresses transcription

The consequences of ZC3H4 recruitment to targets are predicted to be their early termination and subsequent degradation by the exosome, based on the known fate of PROMPTs and eRNAs. To test whether ZC3H4 recruitment can promote these effects, we established a tethered function assay. ZC3H4 was tagged with bacteriophage MS2 coat protein to engineer its recruitment to a reporter containing MS2 hairpin binding sites (MS2hp-IRES-GFP; *Figure 6A*). Importantly, RNA from this reporter is unaffected by endogenous ZC3H4 (*Figure 6—figure supplement 1A*). HCT116 cells were transfected with either of these three constructs together with MS2hp-IRES-GFP and reporter expression assayed by qRT-PCR. Compared to the two controls, tethered ZC3H4-MS2 significantly reduced reporter RNA expression (*Figure 6B*). ZC3H4-MS2 expression does not affect the same reporter lacking MS2 hairpins (*Figure 6—figure supplement 1B*). This directly demonstrates that ZC3H4 recruitment is sufficient to negatively regulate RNA expression, mirroring the upregulation of its endogenous targets seen when it is depleted.

PROMPTs and eRNAs are degraded on chromatin and we wanted to test whether ZC3H4-MS2 affected these nascent RNAs. The reporter experiments above are on total RNA so whether ZC3H4-MS2 exerted its effect at the gene (plasmid) or following its release was uncertain. Therefore, we purified chromatin-associated RNA (*Wuarin and Schibler, 1994*). As mentioned previously, this

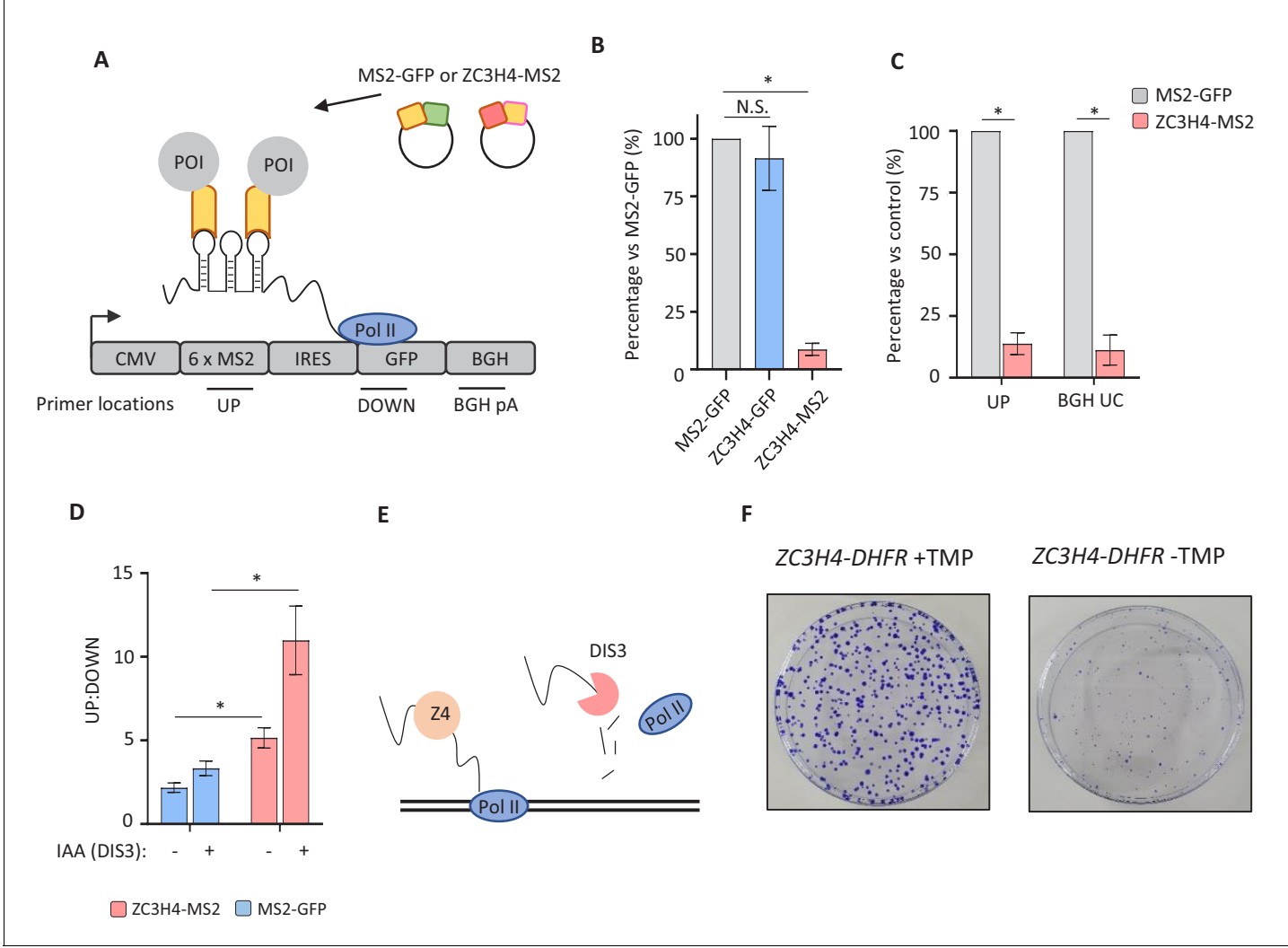

**Figure 6.** Directed recruitment of ZC3H4 recapitulates its effects on endogenous targets. (a) Schematic of the MS2 system. A reporter plasmid (MS2hp-IRES-GFP) expressing a GFP transcript with 6 x MS2 hairpins upstream of an IRES and GFP gene. ZC3H4-MS2 or MS2-GFP can be specifically tethered to the MS2 hairpins to assess consequent effects on transcription/RNA output. Positions of primer pairs used in qRT-PCR experiments elsewhere in the figure are indicated by labelled horizontal lines under reporter. POI is protein of interest. (b) qRT-PCR analysis of total RNA isolated from MS2hp-IRES-GFP transfected cells co-transfected with MS2-GFP, ZC3H4-GFP, or ZC3H4-MS2. The level of reporter RNA is plotted ('UP' amplicon) as a percentage of that obtained in the MS2-GFP sample following normalisation to spliced actin. N = 3. Error bars are SEM. * denotes p<0.05. (c) qRT-PCR analysis of chromatin-associated RNA isolated from MS2hp-IRES-GFP transfected cells co-transfected with either MS2-GFP or ZC3H4-MS2. The level of reporter RNA upstream of the MS2 hairpins (UP) and transcripts yet to be cleaved at the BGH poly(A) site (BGH UC) are plotted as a percentage of that obtained in the MS2-GFP sample following normalisation to spliced actin. N = 3. Error bars are SEM. * denotes p<0.05. (d) qRT-PCR analysis of total RNA isolated from MS2hp-IRES-GFP transfected *DIS3-AID* cells co-transfected with either MS2-GFP or ZC3H4-MS2 – simultaneously treated or not with auxin to deplete DIS3 (14 hr in total). The graph shows the ratio of RNA species recovered upstream (UP) versus downstream (DOWN) of the MS2 hairpins. N = 4. Error bars are SEM. * denotes p<0.05. (e) Schematic detailing an interplay between ZC3H4 and DIS3 that sees transcription stop and nascent RNA degraded (f) Colony formation assay of *ZC3H4-DHFR* cells grown in the presence or absence of TMP. Cells were grown for 10 days before crystal violet staining.

The online version of this article includes the following figure supplement(s) for figure 6:

**Figure supplement 1.** Control experiments for the specificity of ZC3H4 tethering effects.

fractionation enriches nascent endogenous RNAs. However, nascent RNAs associated with transfected plasmids also co-purify within this fraction (*Dye et al., 2006*). Accordingly, cells were transfected with MS2hp-IRES-GFP and either ZC3H4-MS2 or MS2-GFP. We included an additional primer set to detect RNA uncleaved at the bovine growth hormone (BGH) poly(A) site. Because poly(A) site cleavage is co-transcriptional, this primer set should robustly detect Pol II-associated transcripts. This

amplicon and that upstream of the MS2 hairpins were reduced in this chromatin fraction, strongly suggesting that tethered ZC3H4 acts on nascent RNA (*Figure 6C*).

The exosome degrades released PROMPT and eRNA transcripts, which could be enabled by ZC3H4. The results we present for endogenous loci are consistent with this since PROMPTs and eRNAs are upregulated and elongated when ZC3H4 is depleted. To test whether recruited ZC3H4 leads to exosome degradation of RNA, we transfected MS2hp-IRES-GFP, together with either MS2-GFP or ZC3H4-MS2, into *DIS3-AID* cells that were then treated or not with auxin to eliminate the catalytic exosome. RNA upstream and downstream of the MS2 hairpins was detected by qRT-PCR and their ratio plotted (*Figure 6D*). Enhanced levels of upstream versus downstream amplicon were associated with transfection of ZC3H4-MS2 and is more prominent after depletion of DIS3. This is consistent with the hypothesis that recruited ZC3H4 promotes the release of RNA that is a DIS3 substrate (*Figure 6E*). Results presented above show that ZC3H4 functions in transcriptional regulation and it may enable the release of RNA by promoting termination. ZC3H4 may also regulate the stability of its targets; however, to our knowledge, it has not been found to prominently co-purify with the exosome.

Finally, we were interested to determine the overall relevance of ZC3H4 to cell health/growth. This is made simpler by the *ZC3H4-DHFR* cell line, which allows permanent depletion of ZC3H4 by culturing cells without TMP. Accordingly, we performed colony formation assays on these cells grown in the presence or absence of TMP (*Figure 6F*). Loss of ZC3H4 was associated with smaller colonies, which demonstrates the importance of ZC3H4 for growth/proliferation.

## Discussion

We have discovered that ZC3H4 controls unproductive transcription, especially at non-coding loci. This conclusion is based on its recruitment to loci that give rise to transcripts that are stabilised and elongated when it is depleted. Moreover, tethering ZC3H4 to a heterologous reporter RNA is sufficient to promote degradation of the transcript by the exosome. We propose that ZC3H4 recruitment drives some of the early transcriptional termination that is characteristic of many non-coding RNAs, particularly PROMPT and eRNA transcripts. The function of ZC3H4 in restraining their transcription may at least partly explain why PROMPT and eRNA transcripts accumulate as short species when the exosome is depleted.

Our discovery of ZC3H4 adds to an increasing number of termination pathways. Most of these are more relevant during the initial stages of transcription, rather than the more intensively studied process that occurs at the end of protein-coding genes. This is evident from comparing the general requirement for CPSF30 at the 3' end of protein-coding genes with the more selective impact of ZC3H4 that is focused more promoter-proximally. The effects of ZC3H4 depletion are reminiscent of recent findings on the Integrator complex, which also controls the early termination of transcription (*Elrod et al., 2019*; *Lykke-Andersen et al., 2020*; *Tatomer et al., 2019*). Our initial comparison of transcripts sensitive to either Integrator or ZC3H4 suggests that they can act on separate RNA targets. An exciting possibility is that multiple early termination pathways may contribute to conditional gene regulation. It will be important to establish whether ZC3H4 and/or Integrator are naturally utilised to regulate transcription in this manner. Their predominance in metazoans may enable gene regulation, for example across cell types or during development.

ZC3H4 has been proposed as an equivalent to *Drosophila* Suppressor of Sable (Su(s)), which negatively regulates transcription via promoter-proximal termination (*Brewer-Jensen et al., 2016*; *Kuan et al., 2004*). ZC3H4 and Su(s) share little sequence homology, but they have similar structural makeup with zinc fingers flanked by largely disordered regions. Su(s) depletion stabilises selected RNAs and causes their aberrant elongation and stability, mirroring what we see globally following ZC3H4 depletion. There is no known catalytic activity for ZC3H4 or Su(s), but they are related to CPSF30 which shows endonuclease activity in *Drosophila* and *Arabidopsis* (*Addepalli and Hunt, 2007*; *Bai and Tolias, 1996*). It remains to be seen whether ZC3H4 possesses any catalytic activity or mediates its effects through interaction partners. Interestingly, IP and mass spectrometry indicate that WDR82 may be the only interacting partner of Su(s) (*Brewer-Jensen et al., 2016*). WDR82 has been shown to bind to Pol II phosphorylated on Serine 5 of its C-terminal domain, which may provide a means to recruit ZC3H4 to promoter-proximal regions (*Lee and Skalnik, 2008*).

The recruitment of ZC3H4 to promoters is consistent with our observation that promoter-proximal transcription is most affected by its absence. Since depletion of ZC3H4 causes extended transcription of its targets, it is reasonable to suppose that it normally restricts their transcription in some fashion. This might be by controlling the escape of promoter-proximally paused polymerases or by acting closer to the 3′ end of its target transcripts. The fact that ZC3H4 acts somewhat selectively (e.g. not all PROMPTs and mRNAs are its targets) suggests that elements of specificity are required to explain its mechanism. Most obviously, this could be sequences within DNA or RNA, to which ZC3H4 (and Su(s)) binds via ChIP and XRNAX, respectively (see *Figure 5* and *Figure 5—figure supplement 1A*). While our paper was under revision, another report identified ZC3H4 as affecting the transcription of intragenic loci in mammalian cells (*Austenaa et al., 2021*). In agreement with our findings, non-coding transcripts were affected by ZC3H4 depletion. ZC3H4 was proposed to terminate some non-coding transcripts as a result of spurious/weak splicing. Similarly, Su(s) regulation of transcription was linked to the presence of a cryptic 5′ splice site (*Kuan et al., 2004*). This suggests involvement with U1 snRNA, which recognises this sequence. While U1 snRNA inhibition does cause some stabilisation of PROMPTs, it does not generally result in their longer extension and so other *cis*-acting sequences and processes may additionally contribute (*Oh et al., 2017*). Our evidence that ZC3H4 binds RNA in cells suggests that it may directly interact with some of its target transcripts and it will be important to delineate any sequence determinants.

Beyond transcriptional regulation, ZC3H4 occupancy of SEs is interesting. Other notable SE-associated factors (e.g. BRD4 and MED1) are much more generally implicated in Pol II transcription than ZC3H4 (*Sabari et al., 2018*). Moreover, they are transcriptional activators whereas ZC3H4 appears to suppress transcription (or, at least, its RNA output). Many SE-bound factors are found to have phase separation properties explaining why large clusters of factors accumulate at these regions (*Cho et al., 2018*). While we do not know whether ZC3H4 can phase separate, it contains large regions of intrinsic disorder, which can promote this property (*Figure 1—figure supplement 2A*). In general, ZC3H4 may offer a new way to study enhancer clusters, particularly the importance of restricting transcription across these regions.

In conclusion, we have uncovered ZC3H4 as a factor with a function in restricting transcription. Its most notable effects are at non-coding loci where transcriptional termination mechanisms are less understood than at protein-coding genes. Further dissection of ZC3H4 and its targeting should reveal additional important insights into how the unstable portion of the transcriptome is controlled. The non-overlapping effects of Integrator and ZC3H4 at protein-coding genes indicate the possibility that multiple factors may control gene output via premature transcriptional termination.

# Materials and methods

## Key resources table

| Reagent type (species) or resource | Designation | Source or reference | Identifiers | Additional information |
|---|---|---|---|---|
| Cell line (human) | HCT116- CPSF30-mAID | In-house | This paper | |
| Cell line (human) | HCT116- CPSF30-mAID: RPB1-mTurbo | In-house | This paper | |
| Cell line (human) | HCT116- ZC3H4-HA-DHFR | In-house | This paper | |
| Cell line (human) | HCT116- DIS3-AID | In-house | PMID:30840897 | |
| Cell line (human) | HCT116-PNUTS-AID | In-house | This paper | |
| Cell line (human) | HCT116-INTS11-SMASh | In-house | PMID:33113359 | |
| Recombinant DNA reagent | 3xHA-mTurbo-NLS_pCDNA3 | Addgene | RRID #:Addgene_107172 | |
| Recombinant DNA reagent | px300 | Addgene | RRID #:Addgene_42230 | |
| Recombinant DNA reagent | ZC3H4- pcDNA3.1(+)-C-eGFP | Genscript | Custom synthesis | ENTS00000253048 |

*Continued on next page*

*Continued*

| Reagent type (species) or resource | Designation | Source or reference | Identifiers | Additional information |
|---|---|---|---|---|
| Recombinant DNA reagent | pSL-MS2-6x | Addgene | RRID #:Addgene_27118 | |
| Recombinant DNA reagent | pcDNA3.1(+)IRES GFP | Addgene | RRID #:Addgene_51406 | |
| Antibody | CPSF30 | Bethyl | RRID #:AB_2780000 Cat #: A301-585A-T | (1:1000) |
| Antibody | RNA Pol II | Abcam | RRID #:AB_306327 Cat #: ab817 | Now discontinued at abcam (1:1000 for western blot. 4–5 ug used for ChIP qPCR and –seq, respectively) |
| Antibody | PNUTS | Bethyl | RRID #:AB_2779219 Cat #: A300-439A-T | (1:1000) |
| Antibody | WDR82 | Cell Signalling | RRID #:AB_2800319 Clone: D2I3B Cat #: 99715 | (1:1000) |
| Antibody | EXOSC10 | Santa Cruz | RRID #:AB_10990273 Cat #: sc-374595 | (1:2000) |
| Antibody | ZC3H4 | Atlas Antibodies | RRID #:AB_10795495 Cat #: HPA040934 | (1:1000) |
| Antibody | HA tag | Roche | RRID #:AB_390918 Clone: 3f10 Cat #: 11867423001 | (1:2000) |
| Antibody | GFP | Chromotek | Clone: PABG1 Cat #: PABG1-100 RRID #:AB_2749857 | (1:2000) |
| Antibody | TCF4/TCF7L2 | Cell Signalling | RRID #:AB_2199816 Clone: C48H11 Cat #: 2569 | (1:1000) |
| Chemical compound drug | TMP | Sigma | Cat #: T7883 | |
| Chemical compound drug | IAA | Sigma | Cat #: 12886 | |
| Commercial assay, kit | Lipofectamine RNAiMax | Life Technologies | Cat #: 13778075 | |
| Commercial assay, kit | JetPRIME | PolyPlus | Cat #: 114–01 | |
| Commercial assay, kit | Streptavidin Sepharose High Performance slurry | GE Healthcare | Cat #: GE28-9857-38 | |
| Commercial assay, kit | GFP TRAP magnetic agarose | Chromotek | RRID #:AB_2827592 Cat #: gtd-100 | |
| Commercial assay, kit | Dynabeads α-mouse | Life Technologies | RRID #:AB_2783640 Cat #: 11201D | |
| Commercial assay, kit | Dynabeads α-rabbit | Life Technologies | RRID #:AB_2783009 Cat #: 11203D | |
| Commercial assay, kit | SimpleChIP Plus Enzymatic Chromatin kit | Cell Signalling | Cat #: 9005 | |
| Commercial assay, kit | TruSeq Stranded Total RNA Library Prep Kit | Illumina | Cat #: 20020596 | |
| Commercial assay, kit | NEBNext Ultra II DNA Library Prep Kit for Illumina | NEB | Cat #: E7645S | |
| Commercial assay, kit | Ribo-Zero Gold rRNA removal kit | Illumina | Cat #: 20040526 | |
| Commercial assay, kit | Ampure XP beads | Beckman Coulter | Cat #: A63880 | |
| Commercial assay, kit | RNAClean XP Beads | Beckman Coulter | Cat #: A63987 | |

*Continued on next page*

*Continued*

| Reagent type (species) or resource | Designation | Source or reference | Identifiers | Additional information |
|---|---|---|---|---|
| Software, algorithm | BamTools | *Barnett et al., 2011* | RRID #:SCR_015987 | v2.4.0 |
| Software, algorithm | BEDtools | *Quinlan and Hall, 2010* | RRID #:SCR_006646 | v2.26.1 |
| Software, algorithm | Bioconductor | https://bioconductor.org/ | RRID #:SCR_006442 | v3.11 |
| Software, algorithm | DeepTools | *Ramírez et al., 2014* | RRID #:SCR_016366 | v3.3.0 |
| Software, algorithm | DESeq2 | *Love et al., 2014* | RRID #:SCR_015687 | v1.28.1 |
| Software, algorithm | featureCounts | *Liao et al., 2013*; *Liao et al., 2014* | RRID #:SCR_012919 | v2.0.0 |
| Software, algorithm | FIMO | *Grant et al., 2011* | RRID #:SCR_001783 | v5.3.3 |
| Software, algorithm | genomicRanges | http://bioconductor.org/packages/release/bioc/html/GenomicRanges | RRID #:SCR_000025 | v1.40.0 |
| Software, algorithm | ggplot2 | https://cran.r-project.org/web/packages/ggplot2 | RRID #:SCR_014601 | v3.3.3 |
| Software, algorithm | Hisat2 | *Kim et al., 2015* | RRID #:SCR_015530 | v2.1.0 |
| Software, algorithm | IGV | *Robinson et al., 2011* | RRID #:SCR_011793 | v2.8.2 |
| Software, algorithm | MACS2 | *Zhang et al., 2008* | RRID #:SCR_013291 | v2.2.6 |
| Software, algorithm | pheatmap | https://cran.r-project.org/web/packages/pheatmap | RRID #:SCR_016418 | v1.0.12 |
| Software, algorithm | R | https://cran.r-project.org/ | NA | v4.0.4 |
| Software, algorithm | Rstudio | https://rstudio.com/ | RRID #:SCR_000432 | v1.3.1093 |
| Software, algorithm | rtracklayer | https://bioconductor.org/packages/release/bioc/html/rtracklayer | NA | v1.48.0 |
| Software, algorithm | SAMTools | *Li et al., 2009* | RRID #:SCR_002105 | v.1.11 |
| Software, algorithm | Trim_galore! | https://github.com/FelixKrueger/TrimGalore/ | RRID #:SCR_011847 | v.0.6.5dev |

## Cell culture

HCT116 parental cells and engineered cell lines were tested negative for mycoplasma and cultured in Dulbecco modified eagle medium, supplemented with 10% foetal calf serum and penicillin streptomycin (Gibco). For RNAi, 6 or 24-well dishes were transfected with siRNA using Lipofectamine RNAiMax (Life Technologies) following the manufacturers' guidelines. The transfection was repeated 24 hr later and, 48 hr after the second transfection, RNA was isolated. For MS2 assays, cells were seeded in 24-well dishes overnight, then transfected with 50 ng MS2hp-IRES-GFP and 100 ng of MS2-GFP, ZC3H4-MS2 or ZC3H4-GFP using JetPRIME (PolyPlus) for 24 hr. To deplete DIS3-AID or PNUTS-AID, auxin was used at a final concentration of 500 uM. To deplete ZC3H4-DHFR, cells were washed twice in PBS and grown in media with or without TMP (30 uM).

## Cell line generation and cloning

*CPSF30-mAID* and *CPSF30-mAID:RPB1-mTurbo* cells were generated using CRISPR/Cas9-mediated homology-directed repair (HDR). CPSF30 and RPB1 homology arms and gRNA sequences are detailed in *Supplementary file 7*. The mTurbo insert derives from 3xHA-mTurbo-NLS_pCDNA3 (#107172, Addgene). For ZC3H4 degron cells, 3xHA-DHFR was amplified from existing CPSF73-HA-DHFR constructs (published in *Eaton et al., 2018*) using non-homologous end-joining (NHEJ) as described in *Manna et al., 2019*. PNUTS-AID cells were constructed using the protocol described in *Davidson et al., 2019*. In general, 6 cm dishes of cells were transfected with 1 ug of guide RNA expressing px300 plasmid (#42230, Addgene) and 1 ug of each HDR template/NHEJ PCR product. Three days later, cells were seeded, as appropriate, into hygromycin (30 µg/ml, final); neomycin (800 µg/ml, final); or puromycin (1 µg/ml, final). ZC3H4 cDNA was purchased from Genscript in a

pcDNA3.1(+)-C-eGFP vector. The MS2hp-IRES-GFP reporter was made by inserting a BamH1/EcoRV restriction fragment from pSL-MS2-6x (#27118, Addgene) into pcDNA3.1(+)IRES GFP (#51406, Addgene) also digested with BamH1/EcoRV.

## Turbo sample preparation

10 cm dishes at ~80% confluency were labelled with 500 µM biotin for 10 min and the labelling reaction quenched immediately by washing cells in ice cold PBS. Cells were lysed in RIPA buffer (150 mM NaCl, 1% NP40, 0.5% sodium deoxycholate, 0.1% SDS, 50 mM Tris-HCl at pH 8, 5 mM EDTA at pH 8) containing protease inhibitors (cOmplete mini EDTA-free tablets, Roche) for 30 min on ice, then clarified via centrifugation. 350 uL of washed Streptavidin Sepharose High Performance slurry (GE Healthcare) was incubated with biotinylated or control lysates with inversion at room temperature for 1 hr. Samples were then washed twice with RIPA buffer, twice with Urea buffer (2 M urea, 50 mM Tris HCl pH 8), twice with 100 mM sodium carbonate, and once with 20 mM Tris HCl (pH 8), 2 mM $CaCl_2$. Residual final wash buffer was used to resuspend the beads, which were then flash frozen in liquid nitrogen and sent for tandem mass spectrometry at the University of Bristol Proteomics Facility.

## Identifying mass spectrometry candidates

First, contaminant proteins (e.g. keratin) or those that are known to be preferentially biotinylated in ligase assays (e.g. AHNAK) were excluded. Samples with an average abundance ratio $\leq 0.70$ were classified as having a decreased interaction with RNA polymerase II following CPSF30 depletion. Finally, proteins with $\leq 5$ peptides were discarded. Remaining candidates were plotted in *Figure 1F*.

## qRT-PCR

1 µg of total RNA (DNase treated) was reversed transcribed using random hexamers according to manufacturer's instructions (Protoscript II, NEB); cDNA diluted to 50 uL. qPCR was performed using LUNA SYBR (NEB) on a Rotorgene (Qiagen). Fold changes were calculated using Qiagen's software based on delta CT values. Graphs were plotted using Prism (GraphPad). Numbers underpinning qPCR-derived bar graphs are provided in *Source data 1*.

## Antibodies

CPSF30 (A301-585A-T, Bethyl), RNA Pol II (ab817, Abcam), PNUTS (A300-439A-T, Bethyl), WDR82 (D2I3B, Cell Signalling), EXOSC10 (sc-374595, Santa Cruz), ZC3H4 (HPA040934, Atlas Antibodies), HA tag (clone 3f10, 11867423001, Roche), GFP (PABG1, Chromotek), TCF4/TCF7L2 (C48H11, Cell Signalling). Uncropped western blots are provided in *Source data 2*.

## GFP trap

10 cm dishes were transfected (5 ug plasmid, 24 hr), washed with PBS, and lysed for 30 min on ice in 1 mL lysis buffer (150 mM NaCl, 2.5 mM $MgCl_2$, 20 mM Tris HCl pH 7.5, 1% Triton X-100, 250 units Benzonase [Sigma]). Samples were then clarified through centrifugation (12000xg, 10 mins), split in two and incubated with 25 ul of GFP TRAP magnetic agarose (Chromotek) for 1 hr with rotation at 4°C. Beads were washed 5x with lysis buffer and samples eluted by boiling in 2x SDS buffer (100 mM Tris-Cl pH 6.8, 4% (w/v) SDS (sodium dodecyl sulfate), 0.2% (w/v) bromophenol blue, 20% (v/v) glycerol, 200 mM β-mercaptoethanol) before analysis by western blotting.

## Nuclear RNA-Seq

Nuclei were extracted from $1 \times 30$ mm dish of cells per condition using hypotonic lysis buffer (10 mM Tris pH5.5, 10 mM NaCl, 2.5 mM $MgCl_2$, 0.5% NP40) with a 10% sucrose cushion and RNA was isolated using Tri-reagent. Following DNase treatment, RNA was phenol chloroform-extracted and ethanol-precipitated. After assaying quality control using a Tapestation (Agilent), 1 µg RNA was rRNA-depleted using Ribo-Zero Gold rRNA removal kit (Illumina), then cleaned and purified using RNAClean XP Beads (Beckman Coulter). Libraries were prepared using TruSeq Stranded Total RNA Library Prep Kit (Illumina) and purified using Ampure XP beads (Beckman Coulter). A final Tapestation screen was used to determine cDNA fragment size and concentration before pooling and sequencing using Hiseq2500 (Illumina).

## ChIP-qPCR

Cells were cross-linked for 10 min at room temperature (1% formaldehyde) and quenched for 5 mins (125 mM glycine). Cells were washed in PBS, pelleted (500x$g$), and resuspended in 400 ul RIPA buffer (150 mM NaCl, 1% NP40, 0.5% sodium deoxycholate, 0.1% SDS, 50 mM Tris-HCl at pH 8, 5 mM EDTA at pH 8). Sonication was then performed in a Bioruptor (30 s on/30 s off x10 on high setting) and debris pelleted (13000 rpm x 10 min). Supernatants were then incubated for 2 hr at 4°C with 40 ul of sheep anti-mouse dynabeads to which 4 ug of anti-Pol II (or, as a control, nothing) was pre-bound. Beads were washed 6x with RIPA buffer and then bound chromatin was eluted by 30 min incubation at room temperature with rotation (500 ul 0.1 M NaHCO₃ +1% SDS). Cross-links were reversed overnight at 65°C with the addition of 20 ul 5M NaCl. Following phenol chloroform extraction and ethanol precipitation, chromatin was resuspended in 100 ul water of which 1 ul was used per qPCR reaction.

## ChIP-Seq

ChIP libraries were prepared using SimpleChIP Plus Enzymatic Chromatin kit (9005, Cell Signalling) according to manufacturer's instructions. 5 µg of RNA Pol II (abcam, 8WG16) or ZC3H4 (HPA040934, Atlas Antibodies) were used for immunoprecipitation, Dynabeads α-mouse/α-rabbit (Life Technologies) were used for capture.

## Chromatin RNA isolation

HCT116 cells were scraped into PBS, pelleted, incubated in hypotonic lysis buffer (HLB; 10 mM Tris. HCl at pH 7.5, 10 mM NaCl, 2.5 mM MgCl₂, 0.5% NP40), underlayered with 10% sucrose (w/v in HLB) on ice for 5 min, then spun at 500x$g$ to isolate nuclei. Supernatant and sucrose were removed and nuclei resuspended in 100 µL of NUN1 (20 mM Tris-HCl at pH 7.9, 75 mM NaCl, 0.5 mM EDTA, 50% glycerol, 0.85 mM DTT), before being incubated with 1 mL NUN2 (20 mM HEPES at pH 7.6, 1 mM DTT, 7.5 mM MgCl₂, 0.2 mM EDTA. 0.3 M NaCl, 1 M urea, 1% NP40) on ice for 15 min. Samples were spun at 13,000x$g$ to pellet chromatin, this was dissolved in Trizol and RNA extracted.

## Colony formation assays

*ZC3H4-DHFR* cells were seeded into 100 mm dishes and maintained in the presence or absence of TMP for 10 days, with media replaced every 3 days. Colonies were fixed in ice cold methanol for 10 min and then stained with 0.5% crystal violet (in 25% methanol) for 10 min.

## XRNAX

We essentially followed the protocol of *Trendel et al., 2019*. HCT116 cells were grown overnight in the presence or absence of doxycycline in 10 cm dishes. 24 hr later, dishes were washed with PBS, UV cross-linked (Stratalinker 1800 150 mJ/cm2), or not, then resuspended in 4.5 mL Trizol (Sigma). 300 uL of chloroform was added, samples agitated on a ThermoMixer (Eppendorf) for 5 min, spun at 12,000x$g$ for 15 min, then the interphase carefully aspirated into fresh tubes. The interphase was washed thrice with Tris-SDS (10 mM Tris-HCL pH 7.5, 1 mM EDTA, 0.1% SDS), before being dissolved in 1 mL Tris-SDS. 1 uL glycogen, 60 uL of 5M NaCl, and 1 mL isopropanol were added and samples precipitated at −20°C for 10 min, then pelleted at 18,000x$g$ for 15 mins. Precipitated protein was washed with 70% ethanol, air dried, resuspended in 180 uL water and pellets dissolved on ice. DNA was removed via TurboDNase (ThermoFisher) treatment, before samples were repelleted, redissolved in RNase buffer (150 mM NaCl, 20 mM Tris-HCL pH 7.5, 2.5 mM MgCl2) and RNA-digested with RNase A (NEB) and 1 uL of RNase T1 (Roche). 4x SDS loading buffer was added before gel electrophoresis and western blotting.

## Computational analysis

All sequencing data were uploaded to the Galaxy web platform and processed as detailed below; usegalaxy.org and usegalaxy.eu servers were used.

## Datasets (GEO accessions)

Data newly generated in this paper (GSE163015); Pol II HEPG2 ChIP-seq (GSE32883); ZC3H4 HEPG2 ChIP-seq (GSE104247); DIS3-AID HCT116 RNA-seq (GSE120574); INTS1 RNAi Chromatin-

associated RNA-seq (GSE150238). 4sU labelled RNA differential expression in HeLa cells depleted of INTS11 or ZC3H4 (GSE133109, GSE151919).

## RNA-Seq alignment

FASTA files were trimmed using Trim Galore! and mapped to GRCh38 using HISAT2 using default parameters (*Kim et al., 2015*). Reads with a MAPQ score ≤20 were removed from alignment files using SAMtools (*Li et al., 2009*). Finally, BigWig files were generated using DeepTools and visualised using IGV (*Ramírez et al., 2014*).

## ChIP alignment and visualisation

All samples were mapped against GRCh38 using BWA, default settings. Reads with a MAPQ score ≤20 were removed along with PCR duplicates from alignment files using SAMtools. Processed BAM files were converted to BigWig files using DeepTools: all samples were normalised to RPKM with a bin size of 1. Aligned files were visualised using IGV.

## ChIP peak calling

For ZC3H4 and INPUT, broad peaks were called separately using MACS2 with a changed 'lower mfold' (2) and default settings. For each experiment, bedtools was used to establish common peaks from individual reps (Intersect Intervals), creating a bed file of high confidence peaks. For ZC3H4, peaks called in the INPUT sample were subtracted via bedtools. All bed files were annotated and plotted in R using ChipSeeker (*Yu et al., 2015*).

## Gene heat maps

For ChIP heat maps, computematrix (DeepTools) was used to generate score files from ChIP bigwig files using an hg38 bed file; parameters used for each heat map are detailed in figure legends. Plots were redrawn in R. Transcription read-through analysis was calculated for each condition by comparing the first 1 kb downstream of the TES to a 500 bp region directly preceding the TES (PAS). A log2 ratio (depletion/control) was then applied to identify increased read-through.

## SE metaplots

A bed file with the coordinates of SE locations from dbSUPER in HCT116 cells was used as a basis (*Khan and Zhang, 2016*). All regions that had clusters of MED1, Pol II, and H3K27ac ChIP signal were retained as bone fide regions of interest, those without were discarded. A log2 ratio of experiment vs. input was prepared using BamCompare of DeepTools – for RNAseq metaplots, BAM files were split by strand. A score file for the regions in the amended SE bed file was generated via the computematrix function of DeepTools using the log2 BamCompare output file. Results were plotted in R-studio using ggplot2.

## Gene plots and metaplots

Split strand metagene plots were generated using RPKM normalised sense and antisense (scaled to −1) bigwig coverage files separately with further graphical processing performed in R. For identifying ZC3H4 PROMPT regions, ncRNA genes were filtered from hg38 refgene gtf file to give protein-coding genes that were used with feature counts on siCont RNAseq (*Liao et al., 2014*), to gain read count and gene length. Transcripts per million (TPM) were calculated for each gene and genes with an expression <5 were filtered out to give a list of expressed genes. Next, divergent promoters, or genes with neighbours within 5 kb of their promoter, were excluded to minimise background. Finally, this gene list was converted to a bed file, then computematrix (DeepTools) used to generate a score file from log2 siCont Vs condition bigwigs; results were plotted in R.

## Differential gene expression

FeatureCounts was used to count mapped reads over exons and differential expression was performed using DESeq2 (*Liao et al., 2014*; *Love et al., 2014*).

## PROMPT poly-A site detection

For PROMPT analysis, we used a catalogue of 961 PROMPTs generated by de novo assembly following acute DIS3 depletion (*Davidson et al., 2019*). Due to the variable length of each PROMPT, we searched for the two consensus poly-A site motifs (AWTAAA) across the full transcript sequence using FIMO (online). We then calculated the total occurrence of poly-A sites across each PROMPT transcript per kb and separated them into two groups: those that show upregulation (log2FC $\geq$ 1) in the absence of ZC3H4 and those with no change of downregulated expression. Plots were drawn in R.

## ZC3H4 homologue identification

To identify ZC3H4 homologue protein sequences, sequences from UniRef100 (UniProt Consortium, 2014) were searched using a profile HMM search: 'hmmsearch', part of HMMer V3.2.1 (*Eddy, 2011*), with PANTHER (*Mi et al., 2019*) hidden Markov model PTHR13119, corresponding to zinc finger CCCH-domain containing proteins. Profile HMM search hits were filtered using a 1e-100 e-value threshold; this search identified 1513 UniRef100 sequences with PTHR13119 domains (representing a total of 1646 UniProtKB sequences). PTHR13119 domains from human and mouse were aligned using TCoffee Expresso mode (*Armougom et al., 2006*), and multiple sequence alignment figure (*Figure 1—figure supplement 3B*) was rendered with ESPscript (*Robert and Gouet, 2014*).

## Phylogenetic tree reconstruction

Identified PTHR13119 domains were aligned using MAFFT v7.4 *Katoh and Standley, 2013*; sites composed of more than 75% of gaps were removed from the multiple sequence alignment with trimAl *Capella-Gutiérrez et al., 2009*. The PTHR13119 domain phylogeny was reconstructed under maximum likelihood with IQ-TREE v1.6.9 (*Nguyen et al., 2015*). The best-fitting substitution matrix was determined by ModelFinder (*Kalyaanamoorthy et al., 2017*), as implemented in IQ-TREE. Branch support values were based on 1000 ultrafast bootstraps (*Minh et al., 2013*). Phylogenetic Tree figure was rendered with iToL (*Letunic and Bork, 2019*). Multiple sequence alignment and phylogenetic tree files are deposited on Zenodo (https://doi.org/10.5281/zenodo.4637127).

## Primers, siRNAs, and other nucleic acid sequences

Sequences are provided in *Supplementary file 7*.

# Acknowledgements

We are grateful to the other members of the lab for critical comment. This work was supported by a Wellcome Trust Investigator Award (WT107791/Z/15/Z) and a Lister Institute Research Fellowship held by SW. We are grateful to The University of Exeter Sequencing Service where all sequencing was performed who are supported by a Medical Research Council Clinical Infrastructure Award (MR/M008924/1), the Wellcome Trust Institutional Strategic Support Fund (WT097835MF), a Wellcome Trust Multi User Equipment Award (WT101650MA), and a Biotechnology and Biological Sciences Research Council Longer and Larger (LoLa) Award (BB/K003240/1).

# Additional information

### Funding

| Funder | Grant reference number | Author |
| --- | --- | --- |
| Wellcome Trust | WT107791/Z/15/Z | Chris Estell<br>Lee Davidson<br>Steven West |
| Lister Institute of Preventative | | Steven West |
| Royal Society | URF\R1\180537 | Adam Monier |

The funders had no role in study design, data collection and interpretation, or the decision to submit the work for publication.

## Author contributions
Chris Estell, Conceptualization, Formal analysis, Validation, Investigation, Visualization, Writing - original draft; Lee Davidson, Data curation, Investigation, Writing - original draft; Pieter C Steketee, Formal analysis, Visualization; Adam Monier, Formal analysis, Visualization, Writing - original draft; Steven West, Conceptualization, Formal analysis, Supervision, Funding acquisition, Investigation, Methodology, Writing - original draft, Project administration, Writing - review and editing

## Author ORCIDs
Steven West (iD) https://orcid.org/0000-0002-7622-9050

## Decision letter and Author response
Decision letter https://doi.org/10.7554/eLife.67305.sa1
Author response https://doi.org/10.7554/eLife.67305.sa2

# Additional files
## Supplementary files
- Source data 1. Values (average, SEM, p-value) of data underpinning graphs within the paper.
- Source data 2. Uncropped western blot images.
- Supplementary file 1. Mass spectrometry data associated with the Pol II-miniTurbo experiment.
- Supplementary file 2. Underpinning data for phylogenetic analyses.
- Supplementary file 3. Log2 fold changes in read-through following CPSF30 or ZC3H4 depletion from HCT116 cells.
- Supplementary file 4. List of mRNAs that are upregulated following ZC3H4 depletion (nuclear RNA-seq) or INTS1 depletion (chromatin-associated RNA-seq) in HCT116 cells.
- Supplementary file 5. List of mRNAs that are upregulated following ZC3H4 depletion (4sU RNA-seq; *Austenaa et al., 2021*) or INTS11 depletion (4sU TT-seq; *Lykke-Andersen et al., 2020*) in HeLa cells. Genes in each set were manually checked for upregulation (TRUE) or possible artefacts – primarily transcription from surrounding region into a gene that is consequently scored as upregulated (FALSE).
- Supplementary file 6. List of PROMPTs that are upregulated following ZC3H4 depletion (nuclear RNA-seq) or INTS1 depletion (chromatin-associated RNA-seq) in HCT116 cells.
- Supplementary file 7. Table of oligonucleotide and DNA sequences used in this study.
- Transparent reporting form

## Data availability
Sequencing data have been deposited in GEO under accession code GSE163015. All data generated or analysed during this study are included in the manuscript and supporting files. We include full excel spreadsheets representing gene lists and original mass spectrometry data.

The following dataset was generated:

| Author(s) | Year | Dataset title | Dataset URL | Database and Identifier |
|---|---|---|---|---|
| Estell C, Davidson L, Steketee P, Monier A, West S | 2021 | ZC3H4 restricts non-coding transcription in human cells | https://www.ncbi.nlm.nih.gov/geo/query/acc.cgi?acc=GSE163015 | NCBI Gene Expression Omnibus, GSE163015 |

The following previously published datasets were used:

| Author(s) | Year | Dataset title | Dataset URL | Database and Identifier |
|---|---|---|---|---|
| ENCODE | 2011 | Cell-type specific and | https://www.ncbi.nlm. | NCBI Gene |

| | | | | |
|---|---|---|---|---|
| | | combinatorial usage of diverse transcription factors revealed by genome-wide binding studies in multiple human cells | nih.gov/geo/query/acc. cgi?acc=GSE32883 | Expression Omnibus, GSE32883 |
| Partridge EC, Chhetri SB, Myers RM, Mendenhall EM | 2019 | Genome-wide TFBS (Transcription Factor Binding Site) map analysis in HepG2 cells | https://www.ncbi.nlm. nih.gov/geo/query/acc. cgi?acc=GSE104247 | NCBI Gene Expression Omnibus, GSE104247 |
| Davidson L, Francis L, Cordiner RA, Eaton JD, Estell C, Macias S, Caceres JF, West S | 2019 | The immediate impact of exoribonucleolysis on nuclear RNA processing, turnover and transcriptional control revealed by rapid depletion of DIS3, EXOSC10 or XRN2 from human cells | https://www.ncbi.nlm. nih.gov/geo/query/acc. cgi?acc=GSE120574 | NCBI Gene Expression Omnibus, GSE120574 |
| Davidson L, Francis L, Eaton JD, West S | 2020 | An allosteric/intrinsic mechanism supports termination of snRNA transcription. | https://www.ncbi.nlm. nih.gov/geo/query/acc. cgi?acc=GSE150238 | NCBI Gene Expression Omnibus, GSE150238 |
| Lykke-Andersen S, Žumer K, Molska ES, Rouvière JO, Wu G, Demel C, Schwalb B, Schmid M, Cramer P, Jensen TH | 2020 | Integrator is a genome-wide attenuator of non-productive transcription | https://www.ncbi.nlm. nih.gov/geo/query/acc. cgi?acc=GSE151919 | NCBI Gene Expression Omnibus, GSE151919 |
| Austenaa LMI, Piccolo V, Russo M, Prosperini E, Polletti S, Polizzese D, Ghisletti S, Barozzi I, Diaferia GR, Natoli G | 2021 | A first exon termination checkpoint that preferentially suppresses extragenic transcription | https://www.ncbi.nlm. nih.gov/geo/query/acc. cgi?acc=GSE133109 | NCBI Gene Expression Omnibus, GSE133109 |

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
