## [Decision Letter]

Thank you for submitting your article "ZC3H4 restricts non-coding transcription in human cells" for consideration by *eLife*. Your article has been reviewed by 3 peer reviewers, one of whom is a member of our Board of Reviewing Editors, and the evaluation has been overseen by James Manley as the Senior Editor. The following individual involved in review of your submission has agreed to reveal their identity: Joan Steitz (Reviewer #3).

Essential revisions:

The manuscript by Estell et al. investigates the impact of the protein ZC3H4 on RNA levels in human cells. The authors first perform a screen aiming at the identification of factors involved in transcription termination, by identifying proteins in the vicinity of Pol II in a WT context compared to the rapid depletion of the Cpsf30 subunit of the cleavage and polyadenylation complex. The poorly characterised paraloguous proteins ZC3H4 and ZC3H6 display reduced signal upon Cpsf30 depletion. These proteins are then depleted, demonstrating that ZC3H4 depletion gives rise to clear alterations in RNA expression profiles. Based on this, and additional bioinformatics analysis, the authors focus on the impact of ZC3H4 on RNA levels at protein coding genes, PROMPTs and enhancers. It is concluded that ZC3H4 acts as a terminator of transcription for the latter two categories of transcription units. The observed effects are then compared to those of the Integrator complex, that also acts as an early transcription terminator at such loci, which leads to the suggestion that these proteins act on different substrates. Moreover, it is determined that ZC3H4 binds chromatin at regions corresponding to the targeted RNAs. Finally, using a reporter system, the authors further show that ZC3H4 targeting to a transcript reduces its expression level.

The identification of a previously uncharacterized factor involved in transcription regulation is interesting. However, the reported results lack in-depth analysis. Most statements are quite general and would benefit from further analyses.

1. ZC3H4 binds to Pol II in a CPSF30-dependent manner. Moreover, ZC3H4 appears to share similar functions with the Integrator complex. Therefore, the relationship between ZC3H4 and CPSF30 on the one hand and ZC3H4 and Integrator on the other would benefit from further elucidation and exploration. For example:

i. Analyze the connection between ZC3H4 and pA sites e.g by measuring whether ZC3H4-sensitive transcripts are enriched for PASs.

ii. What's the exact number of protein-coding genes demonstrating increased downstream transcription upon depletion of ZC3H4, and how are these distinguished from non-sensitive genes? What is the overlap between CPSF30 and ZC3H4-dependent protein-coding genes?

iii. What are the genes included in the metagene analysis comparing the effects of CPSF30 and ZC3H4 in Figure 3B? Are genes that are read-through in the absence of either CPSF30 or ZC3H4 plotted in the Figure? Or is this a metaplot of the 1795 protein-coding genes demonstrating read-through upon CPSF30 depletion as in Figure 1C?

iv. Comparison of INTS1 and/or ZC3H4-dependent protein-coding genes (related to Figure 4G). Are there any characteristics (e.g. GO processes, gene length, PAS/intron density, expression levels, etc) of the genes upregulated upon depletion of INTS1 or ZC3H4 or both?

v. What is the overlap between the non-coding loci (PROMPTs and SEs) affected by depletion of INTS1 or ZC3H4?

vi. Because the performed screen is related to transcription termination and pA site RNA processing, the described effects of ZC3H4 are predicted to be at the level of transcription termination. However, the possibility that ZC3H4 acts on transcription or RNA stability is not addressed. The usage of the DIS3 depleted samples only serves the purpose of mapping exosome sensitive locations.

2. ZC3H4 versus ZC3H6: Providing a sequence alignment of ZC3H4 and ZC3H6 (or the statistics from such an alignment; %identity/similarity) would yield a better appreciation of the similarity between the two proteins. Overall, this whole panel would be more fitting as a supplementary figure.

3. Figure 4: The comparison of RNA-seq and chromatin-seq is not optimal. This is exemplified in panel E where the two controls display different profiles, leading to a strong effect of Ints1 depletion and no effect of ZC3H4 depletion.

4. Figure 5: While affirming the phenotype also upon rapid depletion of ZC3H4 represents an important control, these data do not bring any new information and should probably be considered supplementary.

5. Figure 6: Considering the weak affinity of the ZC3H4 antibody in ChIP, displaying an input track would help estimate the specificity of the signal. In the same vein: To ensure that the peaks presented in the IGV tracks are not background, one could normalize ChIP-seq data to input DNA and plot the tracks as the ratio of ChIP/Input.

6. Figure 7: While the tethering assay nicely demonstrates that recruitment of ZC3H4 negatively regulates the reporter RNA, additional explanation or experiment is warranted. In their discussion of chromatin-associated RNA, it is unclear how the reporter RNA is found in the chromatin fraction? Do the authors mean chromatin formed on the plasmid or the host chromatin? If the latter, how does the reporter associate with the chromatin? Is detection of uncleaved RNA at the BGH poly A site a faithful readout of nascent RNA? ZC3H4 also affects read-through. Can other methods be used to quantify nascent RNA? The tethering experiment should be performed on at least one endogenous target of ZC3H4 (that was upregulated upon its depletion). Does ZC3H4 over-expression rescue the effect of its depletion or is this not sufficient?

7. Clarification of Figures and Figure legends: Despite presenting interesting experiments, the figures require thorough revision to ensure clarity.

a. Please specify the direction of all genes in the IGV browser tracks in a more visible manner.

b. Adequately label all axes and scales (e.g. Figure 1F, 6A-D, 6G). When referring to fold change, specify of what.

c. When presenting normalized data, include a control in the graph even if its set to 1. (eg. Figure S1C; S3A).

d. Figure 1D: Label CPSF30 green.

e. Figure 2A should be supplementary to Figure 1.

f. Figure 3B: How many genes are being plotted?

g. 3E: Where is MYC SE on the track?

h. Figure 5C and S4: Explain the 5´ and 3´ primer probes (e.g. ITPRID2 5´, ITPRID2 3´).

i. Figure 7D: Both bars should be colored red.

8. Discussion: Several paragraphs in the discussion feel disconnected to the findings presented in the manuscript and how they tie in with the current knowledge regarding transcription termination (eg. Lines 382-395). The authors should revise this section to present a stronger argument highlighting the importance of the findings presented in their manuscript.

---

## [Author Response]

Essential revisions:The manuscript by Estell et al. investigates the impact of the protein ZC3H4 on RNA levels in human cells. The authors first perform a screen aiming at the identification of factors involved in transcription termination, by identifying proteins in the vicinity of Pol II in a WT context compared to the rapid depletion of the Cpsf30 subunit of the cleavage and polyadenylation complex. The poorly characterised paraloguous proteins ZC3H4 and ZC3H6 display reduced signal upon Cpsf30 depletion. These proteins are then depleted, demonstrating that ZC3H4 depletion gives rise to clear alterations in RNA expression profiles. Based on this, and additional bioinformatics analysis, the authors focus on the impact of ZC3H4 on RNA levels at protein coding genes, PROMPTs and enhancers. It is concluded that ZC3H4 acts as a terminator of transcription for the latter two categories of transcription units. The observed effects are then compared to those of the Integrator complex, that also acts as an early transcription terminator at such loci, which leads to the suggestion that these proteins act on different substrates. Moreover, it is determined that ZC3H4 binds chromatin at regions corresponding to the targeted RNAs. Finally, using a reporter system, the authors further show that ZC3H4 targeting to a transcript reduces its expression level.The identification of a previously uncharacterized factor involved in transcription regulation is interesting. However, the reported results lack in-depth analysis. Most statements are quite general and would benefit from further analyses.

We thank all reviewers for taking the time to carefully read our paper and provide the constructive criticism that has helped us to revise it. Below, you will find our detailed responses to each point raised. We would like to point out that while we were revising our paper, another group published some complementary findings (Austenaa et al., 2021 NSMB). They also identify ZC3H4 as a WDR82 interactor and find some defects in intragenic transcription. Our respective independent findings synergise and make a compelling case for the function of ZC3H4 in mammalian non-coding RNA transcription and metabolism.

1. ZC3H4 binds to Pol II in a CPSF30-dependent manner. Moreover, ZC3H4 appears to share similar functions with the Integrator complex. Therefore, the relationship between ZC3H4 and CPSF30 on the one hand and ZC3H4 and Integrator on the other would benefit from further elucidation and exploration. For example:i. Analyze the connection between ZC3H4 and pA sites e.g by measuring whether ZC3H4-sensitive transcripts are enriched for PASs.

Although we identified ZC3H4 via a screen involving CPSF30 depletion, we reported that it did not often affect transcription beyond pA sites of protein-coding genes. This suggests that its transcriptional functions are unlikely to universally require pA sites. As requested, we provide more analyses to support this claim that can be found in Figure 2—figure supplement 2F. We looked at this using PROMPTs since they are easily identified and many are sensitive to ZC3H4 depletion. We took a list of PROMPT transcripts that we previously annotated in HCT116 cells based on their acute sensitivity to exosome (DIS3) loss (Davidson et al., 2019). Of this 961, 438 are upregulated by 2-fold or more following ZC3H4 RNAi. When we compared the pA density in the affected vs unaffected set we found no link between pA density and susceptibility to ZC3H4. In fact, affected PROMPTs tended to have a slightly lower density of pA sequences.

ii. What's the exact number of protein-coding genes demonstrating increased downstream transcription upon depletion of ZC3H4, and how are these distinguished from non-sensitive genes? What is the overlap between CPSF30 and ZC3H4-dependent protein-coding genes?

As we stated in the original manuscript, only a small number of protein-coding genes are affected by ZC3H4 compared to CPSF30 loss. As requested, we now present a bioinformatics analysis (see Figure 2—figure supplement 2A-C). To approximate read-through, we looked at a region 1kb downstream of the poly(A) signal and compared the read density to a region 500bp upstream. This was to maximise the number of genes in the dataset. From over 7000 genes, most show some read-through in the absence of CPSF30 and 4343 have a 2-fold or more increase. By contrast, only 583 have over a 2-fold increase upon ZC3H4 depletion. The majority overlap with those affected by CPSF30 (369/583). In general, the effects of ZC3H4 beyond pA sites are also smaller than for CPSF30. This is illustrated by the fact that the proportion of genes with larger fold effects Log2FC>2 is even lower than those with Log2FC>1 (67 vs 1907 and 583 vs 4343, respectively). Manual inspection of ZC3H4 effects also reveals some read-through examples as having low read coverage beyond the pA site, which may lead to the inclusion of some false positives. However, our analysis provides valuable unbiased support for the generality of CPSF30 effects versus those associated with ZC3H4 loss.

We looked at characteristics of protein-coding genes that show evidence of read-through after ZC3H4 loss. There was no clear effect of length or intron density and only a slight tendency for affected genes to be expressed at lower levels. We have not included these analyses but can do so if necessary.

iii. What are the genes included in the metagene analysis comparing the effects of CPSF30 and ZC3H4 in Figure 3B? Are genes that are read-through in the absence of either CPSF30 or ZC3H4 plotted in the Figure? Or is this a metaplot of the 1795 protein-coding genes demonstrating read-through upon CPSF30 depletion as in Figure 1C?

Genes for these metaplots were not selected based on their sensitivity to ZC3H4 or CPSF30 and were intended to provide unbiased analysis. We selected expressed genes separated from any annotated neighbour by 3kb upstream and 5kb downstream to eliminate closely spaced/overlapping transcription units. The conclusion drawn from them (large/general effect of CPSF30 and small/more selective effect of ZC3H4) is supported by the broader analysis provided in response to the above point (ii) and our extensive manual inspection of the data.

iv. Comparison of INTS1 and/or ZC3H4-dependent protein-coding genes (related to Figure 4G). Are there any characteristics (e.g. GO processes, gene length, PAS/intron density, expression levels, etc) of the genes upregulated upon depletion of INTS1 or ZC3H4 or both?

We looked for enriched GO processes but found nothing striking/obvious in terms of over-represented functions that could link the effects of either factor to large sets of genes with a common function. We can provide tables for this if necessary. Of the other facets, we found that genes affected by INTS1 and ZC3H4 have lower steady-state expression under normal circumstances than those that are unaffected. This is interesting and supports the notion that there are mechanisms (i.e. Integrator and ZC3H4) that restrict their expression. This new data is in Figure 3E.

v. What is the overlap between the non-coding loci (PROMPTs and SEs) affected by depletion of INTS1 or ZC3H4?

The requested analyses are now provided in Figure 3—figure supplement 1C and D. We used our annotated list of 961 PROMPTs, described above for point i), and found that 438 show 2-fold or more upregulation in the absence of ZC3H4 while fewer (222) have this effect without INTS1. 153 of these 222 overlap with those affected by ZC3H4. We had noted in the original manuscript that ZC3H4 effects at PROMPTs were typically greater in magnitude than those associated with INTS1 RNAi. Our new bioinformatics analyses support this. For example, we find that only 26 PROMPTs have a change of 4-fold or more in the absence of INTS1, whereas the figure is 154 for those affected by ZC3H4 depletion. For SEs the effects of INTS1 are much less robust than those of ZC3H4 as can be seen by comparing the metaplots in Figures 2F and 3H.

vi. Because the performed screen is related to transcription termination and pA site RNA processing, the described effects of ZC3H4 are predicted to be at the level of transcription termination. However, the possibility that ZC3H4 acts on transcription or RNA stability is not addressed. The usage of the DIS3 depleted samples only serves the purpose of mapping exosome sensitive locations.

We agree that ZC3H4 effects are predicted to be at the level of transcription (termination). In our view, this is indicated by the long extensions to non-coding RNA caused by its loss. The most credible reason for such distal RNA signal is aberrant transcription caused by ZC3H4 depletion. We say this for two main reasons. First, there is normally no Pol II over these regions indicating that they are not transcribed in unmodified cells. Second, there is no DIS3 effect at these extended regions (only promoter-proximally) arguing against the presence of low level unstable RNA at these positions. To confirm this, we performed Pol II chromatin immunoprecipitation (ChIP) in the presence and absence of ZC3H4 and analysed two loci affected by its loss in our RNA-seq (*MYC* and *ITPRID2* PROMPT regions). This new experiment (Figure 4 —figure supplement 1A) shows increased Pol II occupancy upstream of both PROMPTs following rapid (4 hours) depletion of ZC3H4 and provides important additional support for a transcriptional effect. In addition, we looked at extended transcripts from both PROMPTs via the isolation of chromatin-associated RNA which is enriched in nascent transcripts. This confirmed the ChIP result by demonstrating substantially more (~30 fold) extended RNA following the rapid loss of ZC3H4 (see Figure 4—figure supplement 1B). It is important to note that ZC3H4 may additionally play roles in RNA stability alongside or as part of its function in controlling non-coding transcription. However, to our knowledge, it has never been recovered as an interactor of the exosome complex.

2. ZC3H4 versus ZC3H6: Providing a sequence alignment of ZC3H4 and ZC3H6 (or the statistics from such an alignment; %identity/similarity) would yield a better appreciation of the similarity between the two proteins. Overall, this whole panel would be more fitting as a supplementary figure.

We have now provided the requested alignment and moved the updated figure concerning phylogenetic analyses to the supplement (see Figure 1—figure supplement 3).

3. Figure 4: The comparison of RNA-seq and chromatin-seq is not optimal. This is exemplified in panel E where the two controls display different profiles, leading to a strong effect of Ints1 depletion and no effect of ZC3H4 depletion.

A reason for the two controls looking different is because chromatin-associated RNA (INTS1) is enriched in nascent transcripts whereas the nuclear RNA (ZC3H4) will contain an accumulation of completed mRNA transcripts – hence the higher control level of TM7SF2 versus in the chromatin. We would like to note that not all comparisons look like TM7SF2 with respect to the control profiles. Many examples are not like this because ZC3H4 and Integrator often act on genes that normally show low expression (as we discovered when addressing point 1 iv) above). To better illustrate this, we have provided more IGV snapshots in addition to retaining the TM7SF2 example (Figure 3—figure supplement 1A). We have also provided new data in the revision to further demonstrate that ZC3H4 and Integrator depletion affect different mRNAs and that the ZC3H4 effect is rapid and at chromatin.

To demonstrate that the limited overlap of ZC3H4 and Integrator targets is not due to our comparison of two slightly different sequencing methodologies, we compared recently published datasets that used 4sU to sequence nascent RNA in HeLa cells depleted of the catalytic Integrator subunit (Lykke-Andersen et al., 2020) or ZC3H4 (Austenaa et al., 2021). Similar to our data this revealed several hundred protein-coding transcripts upregulated by either depletion with very little overlap (Figure 3—figure supplement 1B). This orthologous and independent approach, performed in a different cell line, provides strong support for the idea that ZC3H4 and Integrator affect different subsets of mRNAs.

To test the rapidity, and confirm the selectivity, of the ZC3H4 effect in our system, we performed qRT-PCR on chromatin-associated RNA after depletion of ZC3H4-DHFR. We selected three transcripts (NWD1, ENO3 and PJVK) that are affected by ZC3H4 via our nuclear RNA-seq. All three are enhanced in the chromatin-associated fraction after just 4 hours of ZC3H4 depletion (see Figure 4E). The same three transcripts were unaffected by Integrator according to our chromatin-associated RNA-seq – a result that we confirmed by qRT-PCR of chromatin-associated RNA isolated from cells depleted of the catalytic Integrator subunit, INTS11. This experiment demonstrates that at least some transcripts are rapidly and specifically (versus Integrator) upregulated by ZC3H4 loss.

In all, we feel that these new analyses and experiments more convincingly highlight some separate transcriptional functions of ZC3H4 and Integrator at different mRNA subsets.

4. Figure 5: While affirming the phenotype also upon rapid depletion of ZC3H4 represents an important control, these data do not bring any new information and should probably be considered supplementary.

We do agree with this assessment in the context of the original manuscript. However, as part of the revision process we have made more use of the ZC3H4-DHFR cells to address some of the important points raised by the reviewers. E.g. to show that ZC3H4 acts quickly on transcription of mRNA targets and, importantly, that the effects of its depletion are rapidly reversed when its levels are allowed to recover. We have incorporated some of these new data into an expanded version of Figure 5 (now Figure 4).

5. Figure 6: Considering the weak affinity of the ZC3H4 antibody in ChIP, displaying an input track would help estimate the specificity of the signal. In the same vein: To ensure that the peaks presented in the IGV tracks are not background, one could normalize ChIP-seq data to input DNA and plot the tracks as the ratio of ChIP/Input.

As requested, we have included an input track for these experiments. This makes clear that the presented peaks are not background.

6. Figure 7: While the tethering assay nicely demonstrates that recruitment of ZC3H4 negatively regulates the reporter RNA, additional explanation or experiment is warranted. In their discussion of chromatin-associated RNA, it is unclear how the reporter RNA is found in the chromatin fraction? Do the authors mean chromatin formed on the plasmid or the host chromatin? If the latter, how does the reporter associate with the chromatin? Is detection of uncleaved RNA at the BGH poly A site a faithful readout of nascent RNA? ZC3H4 also affects read-through. Can other methods be used to quantify nascent RNA?

We apologise for the lack of clarity here and have provided some more explanation in the revision as requested. As mentioned, chromatin-associated RNA is enriched in nascent transcripts. Importantly, previous reports (e.g. Dye et al., Mol Cell 2006) demonstrated that this method also effectively enriches nascent RNAs transcribed from plasmids. This likely reflects the co-sedimentation of transcribed plasmids with chromatin during this isolation procedure and shows that this method is effective in enriching nascent RNA from exogenous DNA plasmids. Our analysis of RNAs uncleaved at the BGH poly(A) was designed to support this. Because poly(A) site cleavage is co-transcriptional, these transcripts are very likely nascent.

The reviewers also put forward the reasonable opinion that ZC3H4 affects read-through at some protein-coding genes and may therefore influence the BGH pA site to account for some of the tethering effect. We have considered this carefully and are confident that this is not a confounding issue for the following reasons. In response to 1 ii), we show that ZC3H4 rarely affects read-through at pA sites and we have provided (see below) a new experiment that its depletion does not affect the BGH pA site in this reporter. One might posit a scenario in which tethering ZC3H4 to the reporter improves BGH pA processing, thereby accounting for the lower levels of this uncleaved species when ZC3H4-MS2 is used. However, this hypothetical is not supported by our evidence. Such enhanced pA processing would almost certainly result in more reporter mRNA whereas what we observe is much less (i.e. tethered ZC3H4 reduces the recovery of gene body amplicons). Finally, our exosome depletion experiment in this tethering system argues for an effect exerted at, or close to, its site of recruitment (i.e. the MS2 hairpins) and not at the downstream BGH pA. We say this because exosome depletion has a greater effect on the stability of RNA upstream of the MS2 hairpins rather than in the vicinity of the BGH pA.

The tethering experiment should be performed on at least one endogenous target of ZC3H4 (that was upregulated upon its depletion).

We apologise for the lack of clarity here and have provided some more explanation in the revision as requested. As mentioned, chromatin-associated RNA is enriched in nascent transcripts. Importantly, previous reports (e.g. Dye et al., Mol Cell 2006) demonstrated that this method also effectively enriches nascent RNAs transcribed from plasmids. This likely reflects the co-sedimentation of transcribed plasmids with chromatin during this isolation procedure and shows that this method is effective in enriching nascent RNA from exogenous DNA plasmids. Our analysis of RNAs uncleaved at the BGH poly(A) was designed to support this. Because poly(A) site cleavage is co-transcriptional, these transcripts are very likely nascent.

The reviewers also put forward the reasonable opinion that ZC3H4 affects read-through at some protein-coding genes and may therefore influence the BGH pA site to account for some of the tethering effect. We have considered this carefully and are confident that this is not a confounding issue for the following reasons. In response to 1 ii), we show that ZC3H4 rarely affects read-through at pA sites and we have provided (see below) a new experiment that its depletion does not affect the BGH pA site in this reporter. One might posit a scenario in which tethering ZC3H4 to the reporter improves BGH pA processing, thereby accounting for the lower levels of this uncleaved species when ZC3H4-MS2 is used. However, this hypothetical is not supported by our evidence. Such enhanced pA processing would almost certainly result in more reporter mRNA whereas what we observe is much less (i.e. tethered ZC3H4 reduces the recovery of gene body amplicons). Finally, our exosome depletion experiment in this tethering system argues for an effect exerted at, or close to, its site of recruitment (i.e. the MS2 hairpins) and not at the downstream BGH pA. We say this because exosome depletion has a greater effect on the stability of RNA upstream of the MS2 hairpins rather than in the vicinity of the BGH pA.

Does ZC3H4 over-expression rescue the effect of its depletion or is this not sufficient?

We have performed the requested rescue using our *ZC3H4-DHFR* cell line because the degradation of ZC3H4-DHFR that is triggered by withdrawing TMP from media can be reversed by re-administering TMP. This is advantageous to over-expression because re-expression of ZC3H4-DHFR is achieved more quickly than typically required for protein over-expression by transfection. The effectiveness of this approach is shown by a new Western blot in Figure 4C. This re-expression is sufficient to rescue the level of every ZC3H4 target that we tested (see Figure 4D). By supporting our other depletions, the MS2 tethering result and the ChIP showing ZC3H4 occupancy of affected loci, this result provides additional evidence that ZC3H4 is responsible for the effects that we observe in its absence.

7. Clarification of Figures and Figure legends: Despite presenting interesting experiments, the figures require thorough revision to ensure clarity.a. Please specify the direction of all genes in the IGV browser tracks in a more visible manner.

We have now included arrows on each browser track that are colour coded to match sense (blue) or antisense (red) directions.

b. Adequately label all axes and scales (e.g. Figure 1F, 6A-D, 6G). When referring to fold change, specify of what.

We have now done this and provided some additional clarification in the legends.

c. When presenting normalized data, include a control in the graph even if its set to 1. (eg. Figure S1C; S3A).

We have now done this.

d. Figure 1D: Label CPSF30 green.

We have made this change.

e. Figure 2A should be supplementary to Figure 1.

In response to this comment (and another above), we have moved all of the material from the original figure 2 into the supplementary material (Figure 1—figure supplements 2 and 3).

f. Figure 3B: How many genes are being plotted?

As explained above, this is 1795 selected solely on the basis that they are expressed and separated from annotated transcription units (3kb upstream, 5kb downstream). There was no selection for any perceived effect of ZC3H4 and/or CPSF30.

g. 3E: Where is MYC SE on the track?

As explained above, this is 1795 selected solely on the basis that they are expressed and separated from annotated transcription units (3kb upstream, 5kb downstream). There was no selection for any perceived effect of ZC3H4 and/or CPSF30.

h. Figure 5C and S4: Explain the 5´ and 3´ primer probes (e.g. ITPRID2 5´, ITPRID2 3´).

Further information on their location (relative to the beginning of the PROMPT) are provided in the legend for Figure 4D.

i. Figure 7D: Both bars should be colored red.

We have made this change.

8. Discussion: Several paragraphs in the discussion feel disconnected to the findings presented in the manuscript and how they tie in with the current knowledge regarding transcription termination (eg. Lines 382-395). The authors should revise this section to present a stronger argument highlighting the importance of the findings presented in their manuscript.

We have made this change.